# Improving Influence-based Instruction Tuning Data Selection for Balanced Learning of Diverse Capabilities

## Abstract

Selecting appropriate training data is crucial for successful instruction fine-tuning, which aims to (1) elicit strong capabilities from pretrained large language models (LLMs), and (2) achieve balanced performance across a diverse range of tasks. Algorithms based on influence estimation have shown promise in achieving (1) through estimating the contribution of each training example to model's prediction on a downstream task, but they often struggle with (2). Through systematic experiments, we attribute their underperformance to an inherent bias—certain tasks intrinsically have greater influence than others. Directly comparing influence scores across different tasks would thus bias the selected data towards these tasks, hurting the LM's performance not only on other capabilities, but also, surprisingly, on the tasks for which the selected data has high influence.

To address this issue, we propose BIDS, a **B**alanced and **I**nfluential **D**ata **S**election algorithm. BIDS first normalizes influence scores of the training data with respect to each downstream task at an instance level. It then applies an iterative process to further balance the selection of influential training data. At each step, BIDS selects the training example that bears the highest influence on the most underrepresented capability by the currently selected data. We perform comprehensive experiments using both Llama-3 and Mistral-v0.3 on seven evaluation benchmarks spanning five diverse capabilities. Results demonstrate that BIDS consistently outperforms both state-of-the-art influence-based data selection algorithms and other non-influence-based selection frameworks under various budgets. Surprisingly, training on a 15% subset selected by BIDS can even outperform full-dataset training with a much more balanced performance across different tasks. Our analysis further highlights the importance of both instance-level normalization and iterative optimization of selected data for balanced learning of diverse capabilities.

## 1 Introduction

Supervised instruction finetuing (SFT) plays a crucial role in eliciting strong capabilities from large language models (LLMs) and adapting them for various downstream tasks. Typically, a pretrained LLM is finetuned on a mixture of different datasets to achieve strong and balanced performance across a vareity of tasks (Touvron et al., 2023; Dubey et al., 2024; Jiang et al., 2023). Recent efforts have identified the importance of instruction data quality for SFT (Zhou et al., 2024), spawning many works on instruction tuning data selection (Cao et al., 2023; Chen et al., 2023; Liu et al., 2023). Among them, influence-based algorithms have shown promise. They estimate each individual training example's influence on model's prediction on a downstream task (Koh & Liang, 2017; Pruthi et al., 2020). Recent advances in influence estimation methods have enabled their scaling to LLM scales, where they have demonstrated success. These works generally select high-quality data from a large and diverse collection of instruction datasets in order to boost a specific capability of the model (Xia et al., 2024; Yang et al., 2024).

Most real-world applications call for general-purpose LLMs that have both strong and balanced capabilities on a wide range of tasks (e.g., a math tutor that excels in both math problem solving and user instruction following). However, existing influence-based data selection algorithms actually fall short in this aspect (Section 3; Xia et al., 2024). As our analysis reveals, the core of this issue lies in

the fact that when influence-based methods are applied across various tasks, the influence on some of them is inherently higher than others. As a result, directly comparing influence across tasks and selecting examples with highest influence values often result in a selected dataset biased towards the tasks with inherently higher influence. This leads to a couple of pitfalls. First, biasing the selection towards some tasks inevitably comes at the cost of the model's performance on others, making it more challenging for the LLM to achieve balanced capabilities. Second, perhaps unexpectedly, it may even hurt the model's performance on the very task that the data selection is biased towards. These issues call for an influence-based data selection algorithm designed for helping LLMs achieve balanced capabilities across a wide range of diverse tasks through instruction finetuning.

BIDS, our proposed method, addresses these challenges with two key designs. It first normalizes influence values with respect to each validation instance, enabling influence for different instances to be distributed on the same scale. Then BIDS applies an iterative selection algorithm which, at each iteration, selects the training example that provides largest influence improvement for the current selected data. This ensures that each selected example contributes most to the underrepresented tasks in the selected subset.

Our experimental results on two base models of different model families, Llama-3-8B (Dubey et al., 2024) and Mistral-7B-v0.3[1], show the consistent and exceptional effectiveness of BIDS. In our evaluation suite composed of seven tasks that span five diverse capabilities including coding, math, logical inference, world knowledge and general instruction following, BIDS consistently outperforms both influence- and non-influence-based selection algorithms, not only in terms of macro-average performance across diverse tasks, but also in most of the task-specific performance, demonstrating its effectiveness in achieving both balanced and influential data selection. Remarkably, a 15% subset by BIDS even outperforms full-dataset training, emphasizing the huge potential of selective training in fostering multi-capability learning of LLMs. Further analysis reveals the positive contributions from both the instance-level normalization and iterative selection. Investigation of the influence distribution of BIDS-selected data also provides valuable insight on how BIDS reduces the influence disparity across tasks and what might be the good properties of a balanced set of influential data.

We summarize the contribution of this work in the following three aspects:

1. We identify the problem of influence-based data selection methods in instruction tuning LLMs for learning diverse tasks, and attribute this problem to an inherent bias in cross-task influence through systematic analysis.
2. We propose BIDS, a simple and effective influence-based data selection algorithm that selects influential data for balanced capability learning.
3. Through extensive experiments, we confirm the consistent and significant improvement by BIDS, and provide valuable insights on what makes a balanced set of influential data.

## 2 BACKGROUND AND PRELIMINARIES

**Influence-based instruction tuning data selection.** Estimating the influence of individual training examples on model predictions is critical for understanding model behaviors and selecting influential training data to improve model predictions. Traditional methods, including retraining-based and gradient-based approaches (Ilyas et al., 2022; Koh & Liang, 2017), have proven effective but are computationally prohibitive when scaling to LLMs. Several recent advances have sought to address these challenges by extending gradient-based approaches to scale more effectively. Given a large training dataset to select from and a validation set representing some targeted capabilities, LESS (Xia et al., 2024) models the influence between each pair of training and validation example through LoRA-based low-dimensional gradient similarity, and then selects training points with highest influence on the validation set. LOGRA (Choe et al., 2024) leverages a low-rank gradient projection algorithm to further improve the efficiency. MATES (Yu et al., 2024) formulates the pointwise data influence between each training point and the whole validation set, and uses a small data influence model to learn this pointwise influence.

Upon closer investigation, these three LLM-scale influence-based data selection methods all use similar problem formulations. They all need a validation set to represent a targeted data distribution

---

[1] https://huggingface.co/mistralai/Mistral-7B-v0.3

that the selected data are optimized for, and require the computation of pointwise data influence between each training instance and the validation data. In this work, we aim to extend these influence-based methods to a multi-capability instruction tuning setup. Concretely, since only LESS directly targets instruction tuning among the three LLM-scale approaches, we ground our study on the specific formulation of LESS. But we emphasize that due to the highly similar influence modeling patterns shared among these methods, the results of our work should also provide useful insight for other influence-based selection methods.

**Setup.** Below we introduce the problem setup and lay the groundwork for further discussion.

**Definition 1 (Attribution Matrix (AM)).** Assume an instruction tuning dataset $\mathcal{D}$, a validation dataset $\mathcal{V}$, which spans $m$ diverse tasks that we want to optimize the LLM's performance for: $\mathcal{V} = \mathcal{V}_1 \cup \cdots \cup \mathcal{V}_m$, and an **influence estimation method** that can compute the influence of each training example on each validation example. We first compute influences between all training and validation instance pairs, yielding a $|\mathcal{D}| \times |\mathcal{V}|$ matrix $\boldsymbol{A}$. Each row of $\boldsymbol{A}$ corresponds to an individual training example, and each column a validation example. Element $A_{ij}$ indicates the influence of the $i$-th example from $\mathcal{D}$ on the $j$-th example from $\mathcal{V}$. We dub $\boldsymbol{A}$ an **Attribution Matrix** as it reveals the overall attribution pattern from the training set to all target tasks, and each row $\boldsymbol{A}_i$ the **Influence Distribution** of the $i$-th training example.

Our goal is to design a **data selection algorithm** that can effectively select a subset $\mathcal{T}$ from $\mathcal{D}$ with size under a pre-defined budget. based on the influence information in $\boldsymbol{A}$. Finetuning the LLM on $\mathcal{T}$ is supposed to help the model achieve strong and balanced performance on all targeted tasks. Specifically, the tasks are chosen to ensure minimal overlap in terms of the capabilities they evaluate, so that we can maximize the scope of abilities that are evaluated for an LLM. The validation set size for each task is also kept equal to avoid bias in the selection process.

## 3 EXPLORATORY ANALYSIS

**Models, datasets, and tasks.** We use Llama-3-8B (Dubey et al., 2024) as the base model for both influence estimation and evaluation of selected datasets in the exploratory analysis. For the instruction dataset to select from, we use UltraInteract (Yuan et al., 2024), a state-of-the-art, large-scale, high-quality dataset designed to enhance diverse reasoning capabilities, including mathematical reasoning, coding, and general logical inference. We also follow the evaluation setup of Yuan et al. (2024), with seven datasets spanning five diverse capabilities. We use HumanEval (Chen et al., 2021) and MBPP (Austin et al., 2021) for coding, GSM-Plus (Li et al., 2024) and MATH (Hendrycks et al., 2021) for math, and BigBench-Hard (BBH) (Suzgun et al., 2022) for general logical inference. We also use MMLU (Hendrycks et al., 2020) to assess the model's ability to understand and reason over world knowledge, and IFEval (Zhou et al., 2023) for the fine-grained instruction following ability. For more details about the training and evaluation setups, please refer to Appendix A.2.

For the **influence estimation method**, we follow the original pipeline introduced by LESS throughout this work, with an equal number of validation instances sampled uniformly from each of the seven evaluation tasks. In this exploratory analysis, we also adopt the **task-wise max selection algorithm** (Appendix A.3) by LESS, which, for each training example, first computes its mean influence over validation examples within the same task, followed by selecting training examples with the highest maximum influence across different tasks. We compare this algorithm against a random selection baseline, which represents the average performance of models trained on two sets of randomly selected data.

**LESS fails to balance among multiple capabilities.** Table 1 compares LESS against a random selection baseline. Although LESS achieves better task-specific performance in some cases, its overall performance is unbalanced across different tasks, consistently underperforming the random baseline under both the 10% and 15% budgets in terms of macro-average score. LESS also demonstrates significant variability across different tasks. For all budget levels, it significantly lags behind in BBH, while consistently outperforms the random baseline in IFEval.

The unexpectedly low and unbalanced performance of LESS may stem from the fact that it is not designed for learning multiple diverse capabilities, thus less suitable for general-purpose instruction tuning. But the observations above still raise critical questions, especially given that an equal number of validation examples were used for each task during selection. This suggests a potential inherent

Table 1: Comparison between LESS and the random baseline. The highest performance for each task and macro-average is **bolded**. LESS only outperforms the random baseline in macro-average under the 5% budget, while lags behind under both two other budgets with imbalanced performance distribution.

| Budget | Method | Coding | | Logic | STEM | Math | | Ins-Following | Macro Avg |
|---|---|---|---|---|---|---|---|---|---|
| | | HumanEval | MBPP | BBH | MMLU | GSM-Plus | MATH | IFEval | |
| 5% | Random | 43.5 | 48.9 | **64.8** | 64.9 | 41.5 | 22.5 | 18.1 | 43.4 |
| | LESS | **43.9** | **50.7** | 62.7 | **65.1** | **42.5** | **22.6** | **19.7** | **43.9** |
| 10% | Random | **47.8** | 50.6 | **65.0** | **64.9** | 43.9 | 24.0 | 17.8 | **44.9** |
| | LESS | 44.7 | **51.3** | 62.0 | 64.7 | **44.6** | **24.3** | **19.3** | 44.4 |
| 15% | Random | **48.7** | **51.9** | **65.2** | **65.1** | **45.6** | **25.0** | 18.8 | **45.7** |
| | LESS | 46.5 | 51.0 | 63.2 | 64.6 | 44.9 | 24.9 | **21.2** | 45.2 |

bias in the influence values across different tasks, which could skew the selection algorithm towards certain capabilities. If the overall influence on certain task is inherently higher, then the naive task-wise max selection algorithm will naturally prioritize training examples that have high influence on these tasks, possibly at the expense of others. Further, if the oversampling of such training data doesn't bring correspondingly high improvement on this specific task, then the model would not only suffer from an imbalanced learning of the required capabilities, but also a poor overall performance.

To explore this, we need to examine the following two questions: 1) whether influence values differ across tasks and to what extent, and 2) whether higher influence values correlate with greater performance improvements. We aim to answer both questions below.

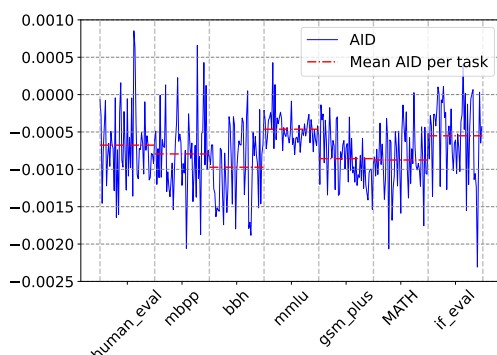

Figure 1: Unnormalized Average Influence Distribution (AID) for all seven tasks under the 10% budget, showing great inter-task and intra-task influence scale differences.

Figure 2: Task frequency with Highest Influence (THI) under the 10% budget. MMLU is obviously oversampled for in LESS-selected data.

**Key takeaways: What causes the imbalanced failure of LESS?**

We first define the following two data analysis techniques to examine the influence distribution of LESS-selected data: for any training subset $\mathcal{T} = \{\boldsymbol{t}_i\}_{i=1}^N$:

- **Average Influence Distribution (AID)**. The AID of $\mathcal{T}$ is defined as the average of influence distributions of all the training points inside, i.e., $\text{AID} \triangleq \frac{1}{N} \sum_{i=1}^N \boldsymbol{A}_i$
- **Task frequency with Highest Influence (THI)**. For each training point $\boldsymbol{t}_i \in \mathcal{T}$, if its average influence on task $j$ is the highest across all tasks, i.e., $j = \arg\max_{k=1,\ldots,m}\{\sum_{\boldsymbol{v}_j \in \mathcal{V}_k} \boldsymbol{A}_{ij}\}$, then the THI frequency for task $j$ increases by one.

Our AID analysis of the whole UltraInteract dataset (Figure 1) reveals both task- and instance-level discrepancies. Notably, MMLU exhibits the highest average influence which is nearly 50% less

negative than BBH, despite neither of them being in-distribution for the training data. Moreover, discrepancies of average influence inside the same task can exceed the largest instance-wise average influence by 2.5 times. These results answer our question 1) by confirming that the scales of influence values indeed differ significantly across various tasks.

Further, the THI analysis of LESS-selected data (Figure 2) validates that the scale differences indeed make the selection algorithm of LESS disproportionately favor certain tasks over others. Specifically, MMLU has the highest frequency of being the most influential task, which is consistent with the observations in Figure 1 that MMLU has the highest task-level average influence. However, the oversampling of MMLU training data does not translate into proportionally better performance—LESS often underperforms the random baseline on MMLU, even though it targets this task with its selected data. This crucial observation underscores that a higher influence score does not necessarily imply a larger performance improvement, and, paradoxically, it may hinder the learning of other necessary capabilities. Thus, we answer the question 2) by concluding that the inherent difference in the influence value scales across tasks is indeed a harmful bias that can severely undermine the performance of the data selection algorithm employed by LESS.

## 4   BIDS: SELECTING INFLUENTIAL DATA FOR BALANCED CAPABILITY LEARNING

To address the imbalanced learning issue caused by the inherent bias in multi-task influence, we propose BIDS, a simple and effective **D**ata **S**election algorithm aiming to select **I**nfluential training data in a **B**alanced way. Given the Attribution Matrix (AM) $\boldsymbol{A}$, BIDS applies two crucial operations to $\boldsymbol{A}$ sequentially: **instance-level normalization**, and **iterative selection favoring underrepresented tasks**.

**Instance-level normalization.** The analyses presented in the above section reveal that validation instances exhibit substantial variability in the scales of their influence values. If left unaddressed, such variability introduces bias into influence-based data selection processes, thus hindering the balanced optimization of model performance across tasks. To mitigate this, we implement an instance-level normalization technique. From the perspective of Attribution Matrix, this technique applies a column-wise normalization in order to align the influence scores on a consistent scale. Specifically, for each validation instance $\boldsymbol{v}_j$, the influence of each training example $\boldsymbol{t}_i$ is normalized as below:

$$\boldsymbol{A}_{ij}^{\mathrm{norm}} = \frac{\boldsymbol{A}_{ij} - \mu_j}{\sigma_j}$$

where $\mu_j$ and $\sigma_j$ are the sample mean and standard deviation of all values in column $j$ of the AM. After such normalization, the influence scores of different columns should be distributed on the same scale, with scores of similar rankings in different columns having similar values.

One potential issue with such a normal standardization technique is that it may not work sufficiently well when the distribution of unnormalized data differs much from an approximate normal distribution. But we show that after normalization, most of the columns in the AM indeed approximate a standard normal distribution, justifying the use of this technique for instance-level normalization of influence scores. Please refer to Appendix A.4 for more details.

**Iterative selection favoring underrepresented tasks.** To further ensure a balanced distribution of influence across the selected data, we propose an iterative greedy selection algorithm (detailed in Algorithm 1). The algorithm begins with an empty set of selected instances and, at each iteration, selects the training instance that provides the largest improvement in influence on any validation instance, ensuring that each newly added instance contributes maximally to underrepresented tasks or capabilities. The iterative process continues until the selection budget is fully utilized. This approach essentially differs from LESS, which only scores each training instance independently and then selects the top-ranked instances, by considering the interactions of influence distributions among different selected instances and promoting the balance of overall influence distribution of the selected dataset.

---

**Algorithm 1** BIDS: Iterative Selection Favoring Underrepresented Tasks

---

1: **Input:** $\mathcal{D}$: a set of training examples, each of which is represented by its normalized influence distribution; $B$: the number of examples to be selected; $\mathcal{V}$: the set of validation examples.
2: **Initialization:** $\mathcal{T} = \varnothing$, $\mathcal{D} = \{\boldsymbol{A}_i\}_{i=1}^{N}$
3: **while** $|\mathcal{T}| < B$ **do**
4: $\quad \boldsymbol{A}^* = \arg\max_{\boldsymbol{A}_i \in \mathcal{D}} \max_{j=1}^{|\mathcal{V}|} \left\{ \boldsymbol{A}_{ij} - \frac{1}{|\mathcal{T}|} \sum_{\boldsymbol{A}_k \in \mathcal{T}} \boldsymbol{A}_{kj} \right\}$
5: $\quad \mathcal{T} = \mathcal{T} \cup \{\boldsymbol{A}^*\}$
6: $\quad \mathcal{D} = \mathcal{D} \setminus \{\boldsymbol{A}^*\}$
7: **end while**
8: **Return:** $\mathcal{T}$: selected training examples.

---

## 5 EXPERIMENTS

### 5.1 EXPERIMENTAL SETUPS

**Basic setup.** We follow the experimental setup outlined in Section 3, including the same set of LLMs, datasets, tasks, and influence estimation implementations. To further validate the generalizability of BIDS, we also perform experiments on base models from different model families. Please find more details in Appendix A.5.

**More Baselines.** For a more thorough comparison, we evaluate BIDS against several other algorithms tailored for influence-based selection, extending beyond the original task-wise max algorithm in the exploratory analysis. Please refer to Appendix A.3 for the mathematical definition of all the influence-based selection algorithms compared in this work. To further show the strength of BIDS among different classes of data selection frameworks, we also compare with a frequently used non-influence-based data selection method. All the additional baselines included in this section are listed below:

- **Instance-wise max.** For each training example, this algorithm first computes the maximum of its influence values over all the validation instances and assigns it as the overall score. Then it selects training examples with highest overall scores.
- **Sum.** Similar to the instance-wise max algorithm, this algorithm also assigns an overall score to each training example and then selects the highest, but by computing the summation of its influence instead of maximum.

- **RDS (Representation-based Data Selection)** (Zhang et al., 2018; Hanawa et al., 2020). We follow the same experimental setup in Xia et al. (2024) and opt for RDS as the non-influence-based baseline to compare with. RDS uses the language model's hidden representation for data selection. More concretely, it extracts the final layer representation of the last token of each example sequence, and computes the cosine similarity scores between training and validation examples. Training examples with the highest similarity to any one of the validation examples are selected. In order to ensure fair comparison, we use the same model that computes gradient features in BIDS to extract the final layer representations for RDS.

### 5.2 MAIN RESULTS

**Comparison under the same budget.** We first compare BIDS with various influence-based selection algorithms as well as RDS under the same budget, as is shown in Table 2. Across multiple budgets spanning 5%, 10% and 15%, we find BIDS consistently outperforms both influence-based algorithms and RDS in terms of the macro-average score across all seven benchmarks. Moreover, when comparing the specific performance on each task separately, we find BIDS ranked either first or second among the six candidate methods on 4/7, 6/7 and 5/7 benchmarks under the three budgets respectively. These two findings together show that BIDS indeed helps improve the LLM's performance on multiple tasks in a significant but also balanced way.

It is also noteworthy that RDS-selected data are significantly biased towards the two coding tasks, HumanEval and MBPP, at the cost of severely degraded performance on almost all other tasks. Specifically, RDS shows the most significant performance drop in math and instruction-following, outperformed by the random baseline on almost all of these results. These observations confirm the

Table 2: Comparison between BIDS and other selection algorithms. The task-specific and macro-average performance is **bolded** if it ranks first under the same budget, and underlined if it ranks second. "BIDS (epochs=4)" is compared with 100% full training. When scaling the training of BIDS to four epochs, it outperforms full-dataset training with both one and four epochs, showing its consistently strong and balanced performance.

| Budget | Method | Coding | | Logic | STEM | Math | | Ins-Following | Macro Avg |
|---|---|---|---|---|---|---|---|---|---|
| | | HumanEval | MBPP | BBH | MMLU | GSM-Plus | MATH | IFEval | |
| 5% | Random | 43.5 | 48.9 | **64.8** | 64.9 | 41.5 | 22.5 | 18.1 | 43.4 |
| | Task-max (LESS) | 43.9 | 50.7 | 62.7 | **65.1** | 42.5 | 22.6 | 19.7 | 43.9 |
| | Sum | **45.6** | 51.9 | 63.6 | 64.8 | 42.4 | 21.3 | 20.1 | 44.2 |
| | Instance-max | 43.9 | 52.1 | 63.2 | 65.0 | **42.6** | 22.3 | 20.6 | 44.2 |
| | RDS | **45.6** | **52.7** | 62.2 | 65.0 | 34.5 | 17.2 | 15.5 | 41.8 |
| | BIDS | **45.6** | 51.0 | 64.3 | 64.9 | 42.1 | **22.9** | **21.4** | **44.6** |
| 10% | Random | 47.8 | 50.6 | 65.0 | 64.9 | 43.9 | 24.0 | 17.8 | 44.9 |
| | Task-max (LESS) | 44.7 | 51.3 | 62.0 | 64.7 | 44.6 | 24.3 | 19.3 | 44.4 |
| | Sum | 45.6 | 51.6 | 61.6 | 64.6 | 43.8 | 23.7 | 21.0 | 44.6 |
| | Instance-max | 46.5 | 47.3 | 64.6 | **65.0** | 44.1 | 24.7 | 22.8 | 45.0 |
| | RDS | **50.0** | **54.7** | 63.2 | 64.6 | 39.3 | 22.4 | 18.3 | 44.6 |
| | BIDS | 48.2 | 50.4 | **65.1** | 64.9 | **45.1** | **25.1** | **23.4** | **46.0** |
| 15% | Random | 48.7 | 51.9 | **65.2** | 65.1 | 45.6 | 25.0 | 18.8 | 45.7 |
| | Task-max (LESS) | 46.5 | 51.0 | 63.2 | 64.6 | 44.9 | 24.9 | 21.2 | 45.2 |
| | Sum | 48.2 | 51.0 | 62.6 | 64.6 | 44.8 | 24.0 | 19.3 | 44.9 |
| | Instance-max | 47.4 | 48.1 | 63.2 | 65.0 | **45.8** | 25.1 | 20.3 | 45.0 |
| | RDS | **50.0** | **53.9** | 63.7 | 64.5 | 41.1 | 23.5 | 18.1 | 45.0 |
| | BIDS | 49.1 | 50.7 | 63.7 | 64.6 | **45.8** | **26.2** | **22.6** | **46.1** |
| | BIDS (epochs=4) | 50.0 | 53.0 | 64.4 | **64.7** | 47.0 | 26.9 | **23.4** | **47.1** |
| 100% | Full (epochs=1) | **52.6** | 53.6 | **65.5** | 64.1 | 47.2 | 27.9 | 17.5 | 46.9 |
| | Full (epochs=4) | 48.2 | **54.4** | 59.2 | 63.1 | **51.5** | **32.3** | 17.9 | 46.7 |

value of further improving influence-based data selection methods in the multi-capability learning setup. More interestingly, they also suggest the possibility that the imbalance of utility scores (Yin & Rush, 2024) may exist for both influence- and non-influence-based data selection approaches. We leave detailed investigations to future work.

**Comparison with full-dataset training.** As shown in the last three rows in Table 2, training on a 15% subset selected by BIDS over four epochs consistently outperforms full-dataset training. Further analysis on task-specific performance reveals that BIDS achieves superior performance by maintaining balanced proficiency across six reasoning-related tasks while significantly improving instruction-following capabilities. These results demonstrate that BIDS not only excels in selecting influential and balanced data, but also that full-dataset training may not always be optimal for fostering robust, multi-capability learning in LLMs. This finding highlights the potential for training on selective subsets to offer more efficient and effective learning effects for LLMs.

# 6  ADDITIONAL ANALYSIS

In this section we mainly discuss the respective contribution of the two additional techniques introduced in BIDS: instance-level normalization, and iterative selection favoring underrepresented tasks. Through extensive ablation experiments and data analysis, we not only validate the respective positive contribution of these two techniques, but also explore how they affect the distribution of selected data specifically, providing valuable insight on what might be the good properties of a balanced set of influential data.

Table 3: Respective contribution of normalization and iterative selection. The highest performance for each task and macro-average is **bolded**. "Unnormalized" refers to algorithm (1), and "Normalized" and BIDS refer to (2) and (3) respectively. It shows both normalization and iterative selection make positive contribution to model performance.

| Budget | Method | Coding | | Logic | STEM | Math | | Ins-Following | Macro Avg |
|---|---|---|---|---|---|---|---|---|---|
| | | HumanEval | MBPP | BBH | MMLU | GSM-Plus | MATH | IFEval | |
| 5% | Unnormalized | 43.9 | **52.1** | 63.2 | **65.0** | **42.6** | 22.3 | 20.6 | 44.2 |
| | Normalized | 45.6 | **52.1** | 62.5 | 64.8 | 42.5 | 22.5 | 20.1 | 44.3 |
| | BIDS | **45.6** | 51.0 | **64.3** | 64.9 | 42.1 | **22.9** | **21.4** | **44.6** |
| 10% | Unnormalized | 46.5 | 47.3 | 64.6 | 65.0 | 44.1 | 24.7 | 22.8 | 45.0 |
| | Normalized | 47.4 | 48.4 | 64.6 | **65.1** | **45.4** | **25.2** | 23.0 | 45.6 |
| | BIDS | **48.2** | **50.4** | 65.1 | 64.9 | 45.1 | 25.1 | **23.4** | **46.0** |
| 15% | Unnormalized | 47.4 | 48.1 | 63.2 | **65.0** | 45.8 | 25.1 | 20.3 | 45.0 |
| | Normalized | 47.4 | 50.1 | **64.9** | **65.0** | 45.6 | 26.0 | 20.8 | 45.7 |
| | BIDS | **49.1** | **50.7** | 63.7 | 64.6 | **45.8** | **26.2** | 22.6 | **46.1** |

## 6.1 BOTH NORMALIZATION AND ITERATIVE SELECTION CONTRIBUTE POSITIVELY

**Algorithms to compare.** In the ablation experiments, we compare the following three selection approaches: (1) Instance-wise max algorithm applied to the original, unnormalized Attribution Matrix computed by LESS; (2) Instance-wise max algorithm applied to the Attribution Matrix after instance-level normalization; (3) BIDS. These three algorithms all originate from the naive instance-wise max approach, but from (1) to (2) the technique of normalization is applied, and from (2) to (3) the iterative selection algorithm is further employed. Therefore, the comparison of these three approaches enables us to clearly see the respective contribution of the two techniques of interest.

**Conclusions.** From Table 3, we observe that instance-level normalization alone can already consistently boost the overall performance of selected data under various selection budgets. And applying the iterative selection not only further elevates the macro-average score, but also improves the balance of model capability across diverse tasks. These two observations confirm that the remarkable performance of BIDS comes from the compound positive effects from both normalization and iterative greedy selection.

## 6.2 EFFECTS OF BIDS ON INFLUENCE DISTRIBUTION OF SELECTED DATA

After confirming the positive contribution from both of the two components of BIDS, we then proceed to explore how they respectively affect the influence distribution of selected data, and whether such effects can provide some insight on why BIDS advances balanced learning of diverse capabilities.

In this section we still compare among the data selected by the three algorithms mentioned in above section: (1) Unnormalized; (2) Normalized; (3) BIDS, based on the two types of data analysis techniques defined in Section 3:

1. **Average Influence Distribution (AID)** across all the 350 validation examples. For better comparison among different algorithms, here we unify the AID analysis with influence values after instance-level normalization.

2. **Task frequency with Highest Influence (THI)** across all seven tasks. Here we use a slightly different definition of task frequency than the one defined in Section 3, by replacing task-wise average influence with instance-wise influence, since the three algorithms we are comparing now are all built upon the instance-wise max approach. Concretely, for each training point $t_i$, if its influence on validation point $v_k$ is the highest across all $|\mathcal{V}|$ validation instances and $v_k \in \mathcal{V}_j$, then the THI frequency for task $j$ increases by one.

Observing the THI analysis results (Figure 3) of these three sets of selected data, we notice that after normalization, the task frequency distribution becomes much more balanced. The frequencies for tasks such as MMLU, GSM-Plus, MATH and IFEval all increase by a great extent, while those

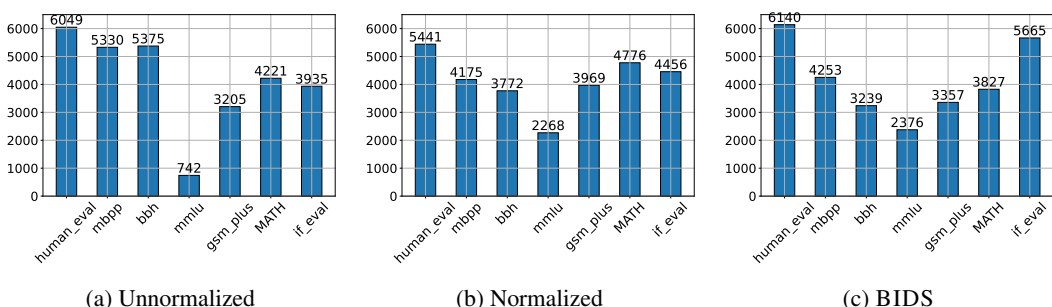

(a) Unnormalized            (b) Normalized            (c) BIDS

Figure 3: Comparative analysis of THI under the 10% budget. Both Normalized and BIDS have more balanced task frequencies compared with Unnormalized.

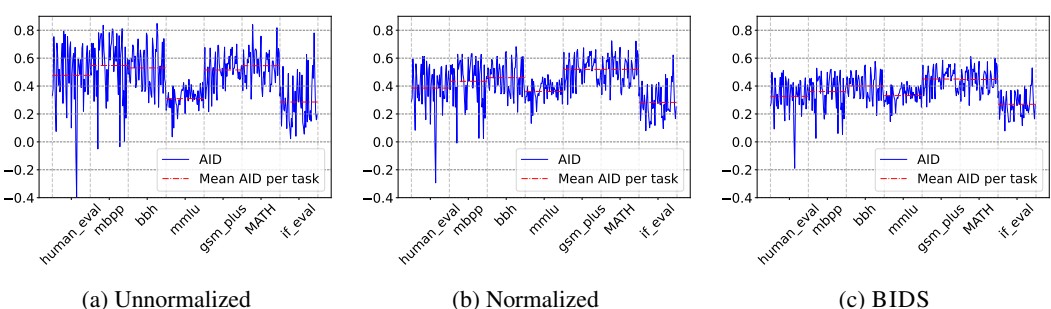

(a) Unnormalized            (b) Normalized            (c) BIDS

Figure 4: Comparative analysis of normalized AID under the 10% budget. From Unnormalized to Normalized to BIDS, the disparity among different tasks and instances in AID gradually diminishes, with both decreasing upper bounds and increasing lower bounds.

for BBH and the two coding tasks decrease. This is fairly surprising when compared with the experimental results in Table 3, where algorithms (2) and (3) actually show improvements both in tasks with decreased and increased THI frequencies compared with (1). This observation suggests that a balanced selection of influential data may improve data efficiency not only by allocating more budget to capabilities that is underrepresented, but also reducing the redundancy in over-represented capabilities.

Additionally, the AID results (Figure 4) provide further insights from a different perspective. From algorithm (1) to (2) to (3), we observe a gradual reduction in the disparity of average influence across tasks. This change unfolds in two interesting ways:

1. The upper bounds of average influence diminish almost for each task. Despite generally lower influence scores across these evaluation tasks, the performance of BIDS improves consistently compared to both the normalized and unnormalized instance-wise max selection algorithms. We remark that this observation actually reveals a limitation of the first-order linearity assumption by the influence estimation method of LESS: simply selecting high-influence points using a top-K algorithm increases the average influence distribution on almost all tasks, but their effectiveness doesn't linearly add up, thus not necessarily improving task-level or overall performance.

2. The lower bounds of average influence increase, especially for tasks with validation instances that have exceptionally low influence values, such as HumanEval and MBPP. This observation again suggests the effectiveness of one of BIDS's key motivations: improving the model's overall performance by enhancing the capabilities that are most underrepresented in the current selected data.

## 7 RELATED WORK

**Data Selection for LLM Instruction Finetuning.** Since the pioneering work LIMA (Zhou et al., 2024) showed that a mere 1000 carefully curated high-quality instruction data can already lead to significant performance improvement, many works have been exploring automatic data selection pipelines guided by different metrics. Quality-guided selection mostly defines the quality for each data point based on natural language indicators (Cao et al., 2023), quality scores from strong evaluators such as GPT-4 (Chen et al., 2023; Parkar et al., 2024), or principled metrics derived from various learning dynamics (Kang et al., 2024; Mekala et al., 2024; Xia et al., 2024; Choe et al., 2024). Diversity-guided methods usually perform clustering over certain informative representation of each data point (Yang et al., 2024), and also take inspiration from traditional core-set selection approaches (Das & Khetan, 2023). Both of these dimensions have been proved effective for instruction tuning LLMs (Bukharin & Zhao, 2023; Liu et al., 2023), and we remark that our method BIDS considers both quality and diversity metrics through its iterative selection algorithm based on influence distributions.

**Influence Estimation.** Influence estimation has long been an important type of data attribution method, which can be classified into gradient-based and retraining-based approaches (Hammoudeh & Lowd, 2024; Ko et al., 2024). Gradient-based influence estimation focuses on the gradient trace of each training point, and assesses the gradient alignment between training and validation examples (Koh & Liang, 2017; Pruthi et al., 2020). Retraining-based estimation usually starts by training models on various subsets, and then inspects how the performance of these models changes when a training example is added to these subsets (Ghorbani & Zou, 2019; Ilyas et al., 2022; Park et al., 2023). Recently both lines of influence estimation works have been extended to LLM-scale applications, covering various aspects including pretraining (Engstrom et al., 2024; Yu et al., 2024; Choe et al., 2024) and instruction tuning (Xia et al., 2024; Liu et al., 2024).

## 8 CONCLUSION

In this work, we introduce BIDS, an influence-based instruction tuning data selection algorithm specifically designed for balanced learning of multiple diverse capabilities. Motivated by the observation of an inherent bias in influence across various tasks, BIDS first applies instance-level normalization to a given Attribution Matrix. Together with an iterative selection algorithm favoring underrepresented tasks, BIDS consistently outperforms various selection algorithms as well as full-dataset training with much more balanced performance. Our analysis further provides insight on the good properties of an influential dataset with balanced capabilities.

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

# A  APPENDIX

## A.1  INFLUENCE ESTIMATION PIPELINE OF LESS

In this section we briefly introduce the influence estimation pipeline of LESS. For more detailed motivation and step-by-step mathematical deduction, we suggest referring to (Xia et al., 2024).

Assume a model $\mathcal{M}_s$ which scores and selects data, and another model $\mathcal{M}_t$ which is trained on the selected data. For a training dataset $\mathcal{D}$ and validation dataset $\mathcal{V}$, LESS formulates the pairwise influence between each training point $t_i \in \mathcal{D}$ and validation point $v_j \in \mathcal{V}$ using the following two steps:

**Step 1: Warmup training with LoRA.** LESS first trains $\mathcal{M}_s$ on a random subset $\mathcal{D}_{\text{warmup}} \subset \mathcal{D}$ for $N$ epochs using LoRA (Hu et al., 2021), checkpointing the model after each epoch to store LoRA parameters $\{\theta_t\}_{t=1}^{N}$.

**Step 2: Gradient computation and projection.** For each checkpoint $\theta_t$ of LoRA-trained $\mathcal{M}_s$, LESS computes the SGD gradient of validation point $v_j$, and further uses random projection (Johnson & Lindenstrauss, 1984; Park et al., 2023) to project the gradient to a tractable lower dimension. The resulting projected gradient is denoted as $\nabla l(v_j; \theta_t)$. LESS also computes and projects the gradient of training point $t_i$, but uses the Adam gradient defined as follows:

$$\Gamma(t_i, \theta_t) \triangleq \frac{m^{t+1}}{\sqrt{v^{t+1} + \epsilon}}$$

where $m^{t+1}$ and $v^{t+1}$ are the first and second moments in the parameter update rule for Adam optimizer.

**Step 3: Gradient matching and influence calculation.** Finally, LESS employs the following cosine-similarity-based approach to calculate the similarity between the gradient of each training and validation example, accumulated over all the warmup training epochs:

$$\text{Inf}_{\text{Adam}}(t_i, v_j) \triangleq \sum_{t=1}^{N} \overline{\eta_t} \cos(\nabla l(v_j; \theta_t), \Gamma(t_i, \theta_t))$$

where $\overline{\eta_t}$ is the average learning rate in the $t$-th epoch.

## A.2  DETAILS OF TRAINING AND EVALUATION SETUPS

Based on the LESS pipeline described above, we further introduce the implementation details of the training and evaluation setups in this work. All the experiments are carried out on 2 80GB H100 GPUs.

**Training Details.** We basically follow the same set of hyperparameters as LESS when training both $M_s$ and $M_t$. Specifically, a batch size of 128 is used throughout all the training processes in this work, along with a learning rate scheduler with linear warm-up, cosine decay, and a peak learning rate of $2 \times 10^{-5}$. For the influence estimation pipeline, we consistently conduct the warmup training of $M_s$ using four epochs and the full training set. For gradient computation and projection, we uniformly sample 50 validation examples from either the validation or the test split (when there is not a separate validation split) of each of the seven evaluation tasks, leading to a total of 350 validation examples. The projection dimension is set as 8192 for all the training and validation instances. For training $M_t$ on the selected data, we consistently train for two epochs if not otherwise specified.

Both the warmup training for influence estimation and the training on selected data are carried out with LoRA. The LoRA configurations are kept the same throughout the experiments, with a rank of 128, an $\alpha$ value of 512, a dropout rate of 0.1, and LoRA matrices being applied to all the attention modules.

**Evaluation Details.** We follow the evaluation convention of UltraInteract (Yuan et al., 2024) by using greedy decoding (i.e., temperature $= 0$) for all the evaluation tasks except for IFEval, where we use temperature $= 0.7$ and take the median result of three random seeds due to the high variability of this task.

## A.3 MATHEMATICAL DEFINITION OF INFLUENCE-BASED SELECTION ALGORITHMS

In this section, we provide the mathematical definition of all the three influence-based selection algorithms that are used in this work. They share the same framework of first assigning an overall influence score $s_i$ to each training example $t_i$ and then selecting examples with the highest scores, and only differ in the specific definition of $s_i$.

**Task-wise Max:** $s_i \triangleq \max_{k=1,\dots,m}\{\sum_{\boldsymbol{v}_j \in \mathcal{V}_k} \boldsymbol{A}_{ij}\}$.

**Instance-wise Max:** $s_i \triangleq \max_{j=1,\dots,|\mathcal{V}|}\{\boldsymbol{A}_{ij}\}$.

**Sum:** $s_i \triangleq \sum_{j=1}^{|\mathcal{V}|} \boldsymbol{A}_{ij}$.

## A.4 EFFECT OF NORMAL STANDARDIZATION ON ATTRIBUTION MATRIX

In this section we aim to justify the application of normal standardization to Attribution Matrix (AM). Specifically, we randomly select five validation instances (i.e., five columns in AM) from each task, and compare their empirical distributions after normalization with a standard normal distribution. The results show that almost all of the columns sampled approximate a standard normal distribution after the instance-level normalization, which justifies the use of normal standardization as the normalization method in BIDS.

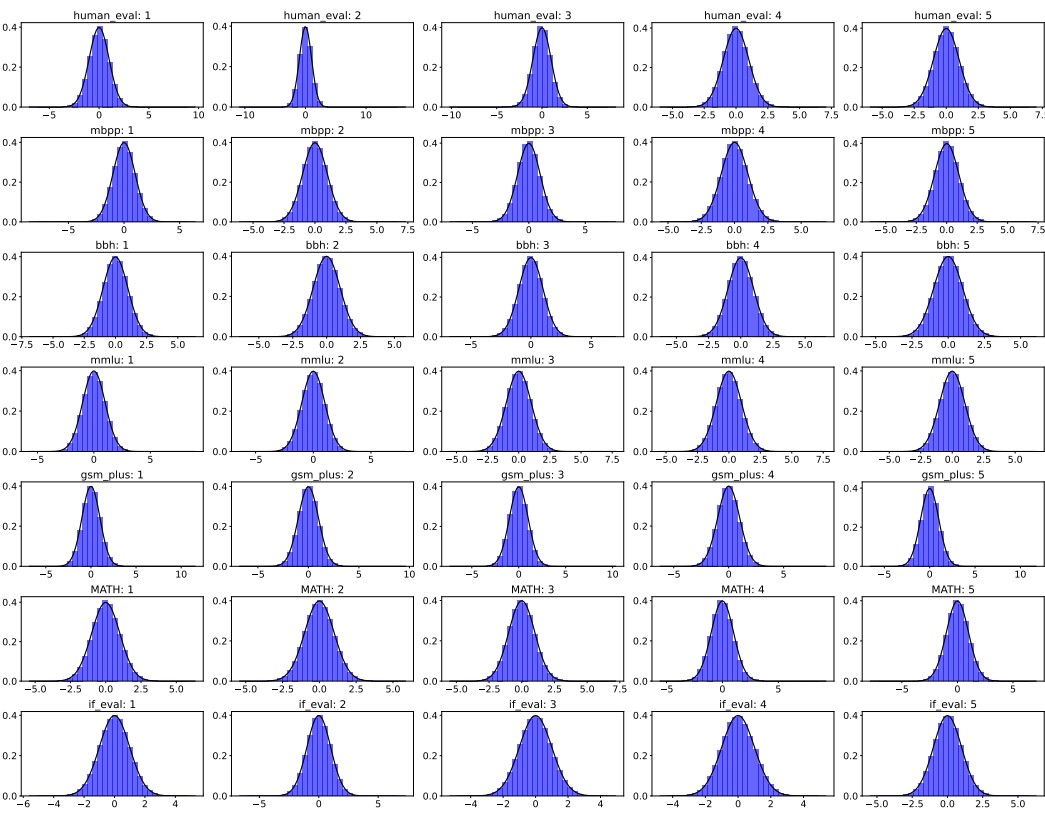

Figure 5: The effect of normal standardization. Five AM columns are sampled for each task. Most of the columns in the AM indeed approximate a standard normal distribution after normal standardization.

## A.5 RESULTS WITH DIFFERENT BASE MODELS

In order to further validate the generalizability of BIDS, we compare BIDS with other baseline data selection algorithms using Mistral-7B-v0.3 as the backbone for both selection and training. The results are presented in Table 4. The two algorithms compared here, Unnormalized and Normalized, follow the same definition in Section 6. And the random baseline is also the average result of two different random seeds.

Table 4: Additional results when using Mistral-7B-v0.3 as the base model for selection and training. The highest performance for each task and macro-average is **bolded**. Under the two selection budgets, BIDS still outperforms all other three baselines with a better macro-avg and more balanced task-specific performance. Also, the performance improvements from Unnormalized to Normalized to BIDS are consistent with prior observation with Llama-3-8B in Section 6. Finally, the top 15% BIDS-selected subset again outperforms full dataset training in macro average, by steadily improving on coding and math while maintaining its remarkable instruction-following ability.

| Budget | Method | Coding | | Logic | STEM | Math | | Ins-Following | Macro Avg |
|---|---|---|---|---|---|---|---|---|---|
| | | HumanEval | MBPP | BBH | MMLU | GSM-Plus | MATH | IFEval | |
| 5% | Random | 36.8 | 44.3 | **59.5** | 61.7 | 37.0 | **19.9** | 22.2 | 40.2 |
| | Unnormalized | 33.3 | **45.0** | 59.3 | 61.6 | 38.0 | 18.7 | 22.0 | 39.7 |
| | Normalized | 36.8 | 44.1 | 59.1 | 61.5 | **38.2** | 19.6 | 27.5 | 41.0 |
| | BIDS | **37.7** | 44.4 | **59.5** | **61.8** | 38.0 | 19.8 | **26.1** | 41.0 |
| 10% | Random | 37.7 | 44.8 | 59.8 | **61.8** | 40.0 | **21.2** | 22.0 | **41.0** |
| | Unnormalized | 36.0 | 43.8 | 59.7 | 61.5 | **41.6** | 20.8 | 24.6 | 41.1 |
| | Normalized | 37.7 | 45.0 | 59.7 | 61.6 | 40.2 | 20.2 | 26.7 | 41.6 |
| | BIDS | **40.4** | **46.1** | **60.5** | 61.7 | 40.5 | 21.0 | **27.1** | **42.5** |
| 15% | BIDS (epochs=4) | 40.4 | 47.0 | **58.9** | **61.1** | 44.1 | 23.5 | **28.1** | **43.3** |
| 100% | Full (epochs=4) | **41.2** | **49.3** | 54.6 | 59.4 | **48.1** | **30.1** | 19.6 | 43.2 |

Similar to the analysis framework in Section 3, we also present the AID analysis of the whole UltraInteract dataset (Figure 6) and the THI analysis of LESS-selected data (Figure 7). Then we follow the workflow in Section 6 to present both the THI and AID analysis for the three progressive algorithms: Unnormalized, Normalized and BIDS (Figure 8, 9). The only difference here is that the selection model is Mistral-7B-v0.3 instead of Llama-3-8B.

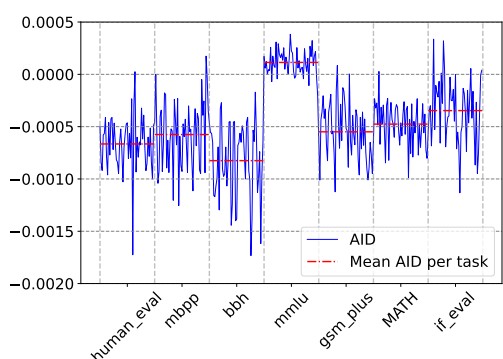 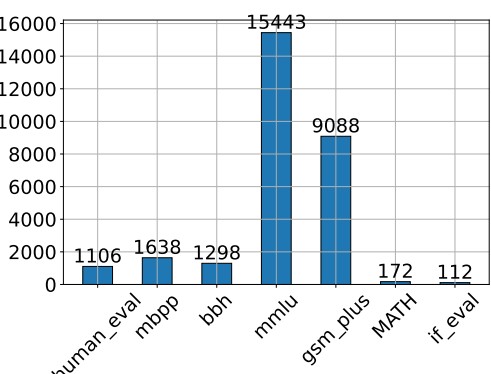

Figure 6: Unnormalized Average Influence Distribution (AID) for all seven tasks under the 10% budget, with the base model being Mistral-7B-v0.3. It still shows great inter-task and intra-task influence scale differences.

Figure 7: Task frequency with Highest Influence (THI) of LESS-selected data under the 10% budget, with the base model being Mistral-7B-v0.3. In this case, MMLU is even more obviously oversampled than prior observation with Llama-3-8B.

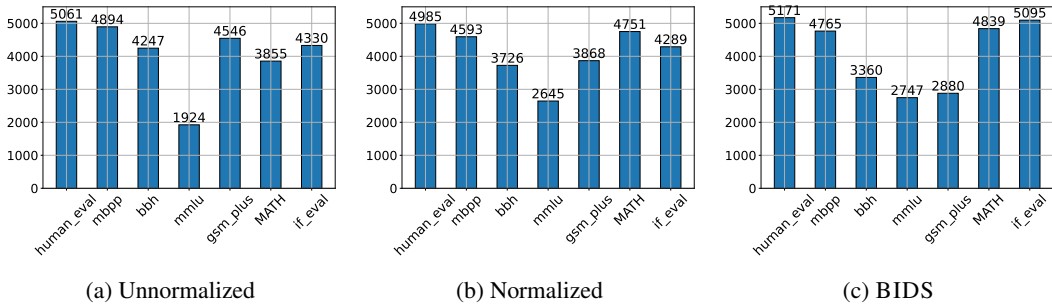

(a) Unnormalized (b) Normalized (c) BIDS

Figure 8: Comparative analysis of THI under the 10% budget, with the base model being Mistral-7B-v0.3. Similar to prior observations with Llama-3-8B, both Normalized and BIDS have more balanced task frequencies compared with Unnormalized.

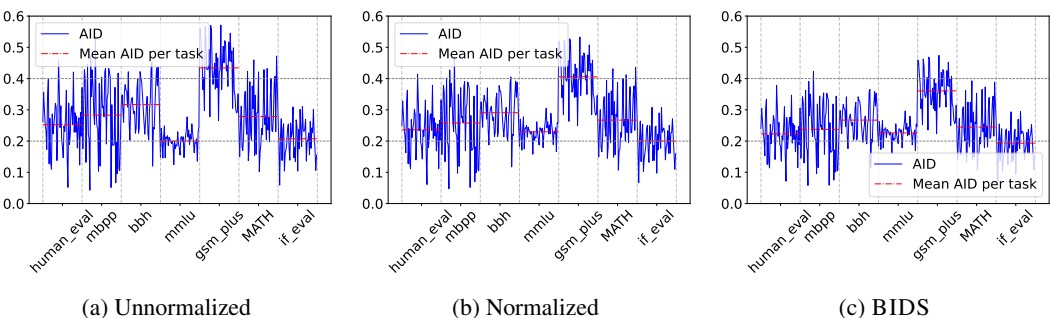

(a) Unnormalized (b) Normalized (c) BIDS

Figure 9: Comparative analysis of normalized AID under the 10% budget, with the base model being Mistral-7B-v0.3. Similar to prior observations with Llama-3-8B, from Unnormalized to Normalized to BIDS, the disparity among different tasks and instances in AID gradually diminishes, with both decreasing upper bounds and increasing lower bounds, although the degree of the original imbalance for Mistral-v0.3 is not as high as Llama-3.

## A.6 DISCUSSION ON THE COMPUTATIONAL COST OF BIDS

In this section we aim to discuss and show that BIDS does not incur much memory or latency overhead, and can thus serve as an efficient plug-and-play module. In our training and evaluation setup, the $|D|$ dimension for the Attribution Matrix (AM) is about 288K, and the $|V|$ dimension is 350. Therefore, the memory cost for storing the AM using FP64 precision is less than 800M. The latency cost for running the whole BIDS algorithm is less than 1 minute with GPU CUDA acceleration. More generally, since most of the popular mixtures of instruction finetuning data are maintained on the scale of hundreds of thousands (Wang et al., 2023; Ivison et al., 2023; Yuan et al., 2024; Yue et al., 2023), the memory and latency cost of BIDS should be light for most practical training setups.

## A.7 QUALITATIVE ANALYSIS

In this section, we aim to show the following two properties of BIDS by providing some qualitative examples:

1. Models trained on BIDS-selected data can indeed achieve a stronger balance between mastering task-specific skills (e.g., math reasoning/coding knowledge, etc.) and fully understanding various types of instructions given by the user (e.g., format-following/response style, etc.).

2. Such a stronger balance is indeed helpful to improving the accuracy or human-perceived quality of model response.

Concretely, we present three sets of model responses in the task of coding (Table 5), math (Table 6) and general instruction-following (Table 7) respectively. Each set contains a correct response by a Mistral-7B-v0.3 model trained on top-15% BIDS-selected data, and a false response by the same base model trained on the full (i.e., 100%) UltraInteract, both to exactly the same prompt. We analyze how the BIDS-trained model correctly answers all these prompts due to the greater balance of capabilities it achieved.

Table 5: For the example 1, the model trained on the full dataset fails to handle the corner case of `numbers = []`. For the example 2, the full-trained model also fails at not adding the constraint of `y != x` in its sorting rule. In both cases, BIDS-trained model returns the correct code completion because it better considers and handles corner cases. It reflects that BIDS-trained model balances its capability in correct coding knowledge and comprehensive thinking behavior.

**HumanEval: Example 1**

**Prompt**

```
1  def rolling_max(numbers):
2      # Write a Python function 'rolling_max(
         numbers: List[int]) -> List[int]' to solve
         the following problem:
3      # From a given list of integers, generate a
         list of rolling maximum element found until
         given moment
4      # in the sequence.
5      # >>> rolling_max([1, 2, 3, 2, 3, 4, 2])
6      # [1, 2, 3, 3, 3, 4, 4]
```

**Wrong Completion: Full-trained model**

```
1  # Initialize the result list with the first
      element of the input list
2  result = [numbers[0]]
3  # Initialize the maximum value with the first
      element of the input list
4  max_value = numbers[0]
5  # Iterate through the input list starting from
      the second element
6  for num in numbers[1:]:
7      # If the current number is greater than the
         maximum value, update the maximum value and
         append the current number to the result list
8      if num > max_value:
9          max_value = num
10         result.append(num)
11     # If the current number is not greater than
         the maximum value, append the maximum value
         to the result list
12     else:
13         result.append(max_value)
14 # Return the result list
15 return result
```

**Correct Completion: BIDS-trained model**

```
1  if not numbers:
2      return []
3  n = len(numbers)
4  result = [numbers[0]]
5  for i in range(1, n):
6      result.append(max(numbers[i], result[-1]))
7  return result
```

**HumanEval: Example 2**

**Prompt**

```
1  def find_closest_elements(numbers: List[float])
      -> Tuple[float, float]
2      # From a supplied list of numbers (of length
       at least two) select and return two that
       are the closest to each other and return
       them in order (smaller number, larger number
       ).
3      # >>> find_closest_elements([1.0, 2.0, 3.0,
       4.0, 5.0, 2.2])
4      # (2.0, 2.2)
5      # >>> find_closest_elements([1.0, 2.0, 3.0,
       4.0, 5.0, 2.0])
6      # (2.0, 2.0)
```

**Wrong Completion: Full-trained model**

```
1  closest_pair = min(numbers, key=lambda x: min(
      abs(x - y) for y in numbers))
2  return closest_pair, min(numbers, key=lambda x:
      abs(x - closest_pair))
```

**Correct Completion: BIDS-trained model**

```
1  # Step 1: Sort the list of numbers in ascending
      order
2  numbers.sort()
3  # Step 2: Initialize the closest_numbers
      variable with the first two elements of the
      sorted list
4  closest_numbers = (numbers[0], numbers[1])
5  # Step 3: Iterate through the list to find the
      two numbers that are the closest to each
      other
6  for i in range(1, len(numbers) - 1):
7      # Step 4: Calculate the difference between
         the current number and the next number
8      diff = abs(numbers[i] - numbers[i + 1])
9      # Step 5: If the difference is less than the
          difference between the current closest
         numbers, update the closest numbers
10     if diff < abs(closest_numbers[0] -
         closest_numbers[1]):
11         closest_numbers = (numbers[i], numbers[i
           + 1])
12 # Step 6: Return the two closest numbers in
      order (smaller number, larger number)
13 return closest_numbers
```

Table 6: For the example 1, BIDS-trained model understands the implicit condition contained in the question that only after a whole number of years can a tree grow oranges for sale, and thus goes on to round the calculated the calculation to integer (Step 7). For the example 2, the model trained on the full dataset fails to understand the meaning of "a person was born 8 years before another" (Step 3) and thus performs the calculation in the wrong way. In both examples, BIDS-trained model correctly answers the question because it balances its capability in math reasoning and natural language understanding.

| GSM-Plus: Example 1 | GSM-Plus: Example 2 |
|---|---|
| **Question** | **Question** |
| Solve the following math problem step-by-step. Simplify your answer as much as possible. Present your final answer as \boxed{Your Answer}. Carlos is planting an orange tree. The tree will cost $120 to plant. Each year it will grow 10 oranges, which he can sell for $2 each. It costs $4 a year to water and feed the tree. How many years will it take before he starts earning money on the orange tree? | Solve the following math problem step-by-step. Simplify your answer as much as possible. Present your final answer as \boxed{Your Answer}. Raymond and Samantha are cousins. Raymond was born 8 years before Samantha. Raymond had a son at the age of 25. If Samantha is now 35, how many years ago was Raymond's son born? |
| **Wrong Response: Full-trained model** | **Wrong Response: Full-trained model** |
| Step 1: To determine the number of years it will take for Carlos to start earning money from the orange tree, we need to consider the initial cost of planting the tree and the annual expenses associated with watering and feeding it. Step 2: The tree costs $120 to plant, and it incurs an annual expense of $4 for watering and feeding. Step 3: Each year, the tree produces 10 oranges, which he can sell for $2 each. Step 4: So, the total income from selling the oranges each year is 10 * $2 = $20. Step 5: However, we need to subtract the annual expenses of $4 from the income to find the net profit from the oranges. Step 6: Therefore, the net profit from the oranges each year is $20 - $4 = $16. Step 7: To determine the number of years it will take for Carlos to start earning money, we can divide the initial cost of planting the tree by the net profit from the oranges each year. Step 8: This gives us the number of years it will take for the tree to start generating a positive return on investment. Step 9: So, the number of years it will take before Carlos starts earning money on the orange tree is $120 / $16 = 7.5$ years. Answer: \boxed{7.5} | Step 1: To solve this problem, we need to first understand the given information and then work through the steps to find the age of Raymond's son. Step 2: We're given that Raymond was born 8 years before Samantha, and Samantha is currently 35 years old. Step 3: This means that Raymond was born 35 - 8 = 27 years ago. Step 4: Now, we know that Raymond had a son at the age of 25. So, we need to find out how many years ago Raymond's son was born. Step 5: To do this, we can simply subtract Raymond's age when his son was born (27 years) from the current age of Samantha (35 years). Step 6: This gives us 35 - 27 = 8 years. Step 7: Therefore, Raymond's son was born 8 years ago. Answer: \boxed{8} |
| **Correct Response: BIDS-trained model** | **Correct Response: BIDS-trained model** |
| Step 1: To determine how many years it will take for Carlos to start earning money from his orange tree, we need to consider the costs involved in planting and maintaining the tree, as well as the revenue generated from selling the oranges. Step 2: First, we need to calculate the total cost of planting the tree, which is $120. Step 3: Each year, the tree will grow 10 oranges that can be sold for $2 each, resulting in a total revenue of $20 per year. Step 4: However, it also costs $4 per year to water and feed the tree. Step 5: Therefore, the net revenue from selling the oranges each year is $20 - $4 = $16. Step 6: To determine how many years it will take for Carlos to start earning money, we can divide the total cost of planting the tree by the net revenue each year. This gives us 120 / 16 = 7.5 years. Step 7: Since we can't plant a tree for half a year, Carlos will need to wait 8 years before he starts earning money from the orange tree. Answer: \boxed{8} | Step 1: To solve this problem, we need to understand the relationship between Raymond and Samantha, and then calculate the age of Raymond's son based on the given information. Step 2: Raymond is Samantha's cousin, and he was born 8 years before her. This means that Raymond is 8 years older than Samantha. Step 3: Raymond had a son at the age of 25, so we need to calculate Raymond's age at that time. Since Raymond is 8 years older than Samantha, and Samantha is currently 35, we can calculate Raymond's age as 35 + 8 = 43. Step 4: Now, we know that Raymond had his son at the age of 25, so we can calculate the age of Raymond's son by subtracting 25 from Raymond's age at that time, which is 43 - 25 = 18. Step 5: Therefore, Raymond's son was born 18 years ago. Answer: \boxed{18} |

Table 7: For the example 1, the model trained on the full dataset ignores the explicit requirement of forming its output as a single JSON block. It is especially overfit to the response format of `Step X: ...` which is frequently seen in the training data. In contrast, BIDS-trained model handles its CoT in a nifty way by expressing its thinking process in the values of a JSON block. For the example 2, full-trained model not only fails to completely follow the explicitly required format, but also hardly writes a poem, compared with the high-quality one generated by BIDS-trained model. In both examples, BIDS-trained model generates content with significantly higher quality due to the balance between strict format-following and flexible use of parametric knowledge.

| **IFEval: Example 1** | **IFEval: Example 2** |
|---|---|
| **Prompt** | **Prompt** |
| Please provide the names of 5 famous moms in JSON format. Please, use any interesting or weird tone. Your entire output should just contain a JSON block, nothing else. | Write a poem about how I am missing my classes. The poem must have 4 sections marked with SECTION X. Finish the poem with this exact phrase: "Can I get my money back for the classes I missed?" |
| **Wrong Response: Full-trained model** | **Wrong Response: Full-trained model** |
| Step 1: Let's think of 5 famous moms who have made a significant impact in their respective fields.
Step 2: Oprah Winfrey - Talk show host, media mogul, producer, actress, philanthropist
Step 3: Angelina Jolie - Actress, filmmaker, humanitarian
Step 4: Michelle Obama - Former First Lady of the United States
Step 5: Serena Williams - Professional tennis player
Step 6: Create a JSON block with the names of the 5 famous moms. | Section 1:
I missed my first class today and I'm feeling quite disappointed.
Section 2:
I really wanted to learn all the new material, but I had to stay home due to illness.
Section 3:
I hope I can make up for the missed class by studying extra hard for the next exam.
Section 4:
Can I get my money back for the classes I missed? |
| ```javascript
// Step 7: Create a JSON block with the names of the 5 famous moms.
[
"Oprah Winfrey",
"Angelina Jolie",
"Michelle Obama",
"Serena Williams"
]
``` | Answer:
"I missed my first class today and I'm feeling quite disappointed.
I really wanted to learn all the new material, but I had to stay home due to illness.
I hope I can make up for the missed class by studying extra hard for the next exam.
Can I get my money back for the classes I missed?" |
| **Correct Response: BIDS-trained model** | **Correct Response: BIDS-trained model** |
| {
"Meryl Streep": "An American actress and filmmaker, known for her versatile and powerful performances in films like Kramer vs. Kramer, Sophie's Choice, The Devil Wears Prada, and many more.",
"Oprah Winfrey": "An American media mogul, talk show host, actress, producer, and philanthropist. She is best known for her talk show The Oprah Winfrey Show, which ran for 25 years.",
"Sarah Jessica Parker": "An American actress, producer, and designer. She is best known for her role as Carrie Bradshaw in the HBO series Sex and the City.",
"Jennifer Garner": "An American actress, producer, and entrepreneur. She is best known for her roles in films like 13 Going on 30, Juno, and Alias.",
"Angelina Jolie": "An American actress, filmmaker, and humanitarian. She is best known for her work as a United Nations Special Envoy and as a Goodwill Ambassador."
} | SECTION 1:
I'm sitting here feeling so lonely,
Missing my classes with all my friends,
Wishing I could be there with them,
Instead of sitting at home and mend.
SECTION 2:
I'm looking at my empty calendar,
Wondering what I'm missing out on,
Wishing I could join in on the fun,
Instead of staring at the screen and frown.
SECTION 3:
I'm listening to the sound of silence,
While my friends are making memories,
Wishing I could join in on the laughter,
Instead of feeling left out and sadder.
SECTION 4:
I'm staring at the clock, counting the minutes,
Until I can see my teachers and classmates,
Wishing I could just teleport there,
Instead of waiting for the next class to start.

Finish:
Can I get my money back for the classes I missed? |

