# OpenReview forum: "Improving Influence-based Instruction Tuning Data Selection for Balanced Learning of Diverse Capabilities"
_ICLR.cc/2025/Conference — ICLR 2025 Conference Withdrawn Submission_

### Official Review · Reviewer_XLPB · 2024-10-28

**Soundness:** 2
**Presentation:** 2
**Contribution:** 1
**Rating:** 3
**Confidence:** 4

**Summary:**

The paper proposes BIDS, a data-selection algorithm designed to address the inherent bias in existing influence-based data selection methods for instruction tuning of large language models. Build upon the previous work LESS, they first apply instance-level normalization on the influence scores, then utilize an iterative selection algorithm to identify influential samples while maintaining a balance between tasks.

**Strengths:**

- Overall, the writing is clear and easy to follow, and the problem setup is well-defined.
- The paper provides a detailed analysis of the limitations of previous work.
- Experimental results demonstrates the propose method's effectiveness across the selected baselines (however, see weaknesses).

**Weaknesses:**

- This proposed is somewhat incremental, with limited technical contributions. It heavily relies on the previous work LESS, with the primary contributions being instance-level normalization and an “iterative” selection algorithm. However, normalization is already well-explored in ML, and the iterative selection is very similar to “maximal marginal relevance"[1].

  [1] Carbonell, Jaime, and Jade Goldstein. "The use of MMR, diversity-based reranking for reordering documents and producing summaries." *Proceedings of the 21st annual international ACM SIGIR conference on Research and development in information retrieval*. 1998.

- Experiments are insufficient. There is no direct comparison between the proposed method and its previous work, LESS, or other data selection methods. The selected baselines are naive. Additionally, experiments are limited to LLaMA-3-8B, leaving the generalization capacity of the proposed method unproven.

- The experimental results are not very convincing. The performance of the proposed method is worse than the random baseline on some benchmarks, and there is no clear clarification or explanation in the paper. Besides, there are some presentation issues in table 2. It is misleading to "bold" the results if it outperforms two baselines (even if it is specified in the caption). In addition, results for the 5% budget and 15% budget on MMLU should not be bolded.

**Questions:**

- Why do you call it "instance-level" normalization? Can you provide some more comparisons between different normalization methods?
- Can you provide additional results on different LLMs (e.g. Mistral-7B, LLama-2-7B/13B as used in LESS)? It might better demonstrates the generalization of the proposed method.
- Can you provide some examples (LLM response) to show that the proposed method is better for balancing across tasks?

---

> ### Author Response · Authors · 2024-11-22
> **Response to Reviewer XLPB (1/1)**
>
> Thank you for your detailed feedback! We are encouraged that you acknowledge both the effectiveness and the analysis of our method. We address your questions and concerns below.
>
> ## 1. Limited technical contribution of BIDS due to heavy reliance on LESS and use of well-explored techniques
> We sincerely appreciate the references to prior ML research provided by the reviewer. We kindly suggest referring to the general response, **2. Elaboration on the novelty and contribution of BIDS**, for more details on the contribution and novelty of BIDS.
>
> ## 2. Additional results with different base models
> Please also refer to the general response, **1. Validating the generalizability of BIDS with different base models**, where we provide a comprehensive evaluation of BIDS on Mistral-7B-v0.3.
>
> ## 3. Comparison with both LESS and other data selection methods
> First, we point out that in Table 2 of the original main text, we compare our method, BIDS, with the prior work LESS. The method name for LESS is denoted as “Task-max”, since LESS employs a task-wise maximum selection algorithm as is detailed in Section 3. For better clarification we have updated the explicit name of LESS directly into the method name column of Table 2.
>
> Next, as for comparison with other data selection methods, please also refer to the general response, **3. Comparison with non-influence-based data selection baseline**.
>
> ## 4. The experimental results are not very convincing
> We first thank the reviewer for pointing out presentation issues in Table 2. To improve clarification, we have removed the “**bolded + underlined**” notation. Instead, we only **bold** the results if they rank first under the same budget, and **underline** them if they rank second. Next, we explain why the performance of BIDS is worse than random baseline in certain benchmarks.
>
> 1. As is stated in prior work [1, 2], for LLM instruction tuning, the inclusion of certain training data would hinder the learning of other capabilities. In other words, empirically there isn’t a dataset that performs optimally for all tasks. Though BIDS is specifically designed to address this issue, and it indeed achieves consistently better macro average performance across all seven tasks, it’s still normal that BIDS may occasionally lag behind the random baseline in certain task(s), due to these tasks’ conflicting nature.
> 2. Under the budget of both 5% and 10%, BIDS only slightly lags behind the random baseline in one task (0.5% in BBH and 0.2% in MBPP, respectively), while significantly outperforms the latter in multiple other tasks as well as the macro average. Under the 15% budget, though BIDS is outperformed by the random baseline in 3/7 tasks, it still maintains its superior performance in macro average, and beats all the other data selection baselines. It’s also reasonable that with the increase of data selection budget, the performance of random selection would gradually converge to more complicated selection methods.
> 3. Moreover, compared with other influence-based data selection algorithms, BIDS achieves superior performance with only minimum additional memory or latency cost, further demonstrating its efficiency and value. Please refer to Appendix A.6 of the updated manuscript for relevant details.
>
> ## 5. The difference between “instance-level” and “task-level” normalization
> These two terms differ in how they apply normalization to the attribution matrix (AM). Instance-level normalization is applied to each column of the AM by calculating the column mean and standard deviation, while the task-level one is applied to all the columns belonging to the same validation task. The reason why we choose instance-level normalization is that according to Figure 1 in the main text, even the discrepancies of average influence on different validation examples (i.e., columns) in the same task are significant. So we hypothesize that an instance-level normalization could better align the influence on different validation examples on the same scale, thus paving the way for more balanced data selection.
>
> ## 6. Qualitative examples to show the benefits of balanced data selection
> We appreciate the reviewer for proposing this great suggestion. We have collected three sets of model responses in the task of coding, math and general instruction-following respectively. Each set contains a correct response by a BIDS-trained model and a wrong response by a full dataset-trained model, both to exactly the same prompt. In Appendix A.7, we have updated these examples and detailed analysis of how the BIDS-trained model correctly answers all these prompts due to the greater balance of capabilities it achieved.
>
> [1] Wang et al., How Far Can Camels Go? Exploring the State of Instruction Tuning on Open Resources, 2023
>
> [2] Ivison et al., Camels in a Changing Climate: Enhancing LM Adaptation with Tulu 2, 2023

---

> ### Author Response · Authors · 2024-11-28
> **Looking forward to further discussions**
>
> Dear Reviewer XLPB,
>
> Thank you for your valuable feedback on our submission! We have attempted to respond to your comments in our rebuttal, and tried to address your concerns with additional experiments and elaboration. We would greatly appreciate any further clarifications or discussions to enhance our work.
>
> Look forward to further discussions, and thanks again for your time and efforts!

---

> ### Comment · Reviewer_XLPB · 2024-11-28
>
> Thanks the authors for the rebuttal.
>
> However, the authors do not adequately address my concern regarding the novelty or contribution of their work. I still find the paper heavily reliant on prior work LESS, with the proposed improvements appearing incremental. From the reviewer's side, the paper primarily applies well-established techniques that have demonstrated effectiveness in the past, but without providing substantial new scientific insights or in-depth analysis.
> - It is not convincing to say "our findings on the limitations of previous works are novel". Besides, it is unclear how the proposed analysis framework introduces any significant novelty.
> - Although simplicity can indeed be a valuable trait, the rebuttal does not convincingly establish why this specific implementation of simple techniques constitutes a meaningful contribution or provides enough technical insight. e.g. The proposed method appears very similar to the MMR algorithm. How does your method differ from it, and what unique insights or contributions does it offer?
>
> Besides, while the authors claimed in Appendix A.6 that the cost of BIDS is light, there is no direct numerical comparison of memory and computational cost with other methods. E.g. Data selection time using the same hardware (CPU and GPU)? Exact memory cost?

---

> ### Author Response · Authors · 2024-12-04
> **Further Response to Reviewer XLPB (1/2)**
>
> We sincerely thank the reviewer for this reply. Here we try to further elaborate on the novelty of our work compared with the MMR algorithm, and the efficiency comparison with other data selection methods.
>
> ## 1. Novelty compared with prior work
> First, we regard our work as novel since to the best of our knowledge, it is the first to formally raise and address the imbalanced influence issue when applying influence-based data selection methods to multi-task learning. We provide solid analysis on the identification and confirmation of this problem in Section 3, and show it consistently and significantly degrades the performance of selected data across different tasks and models in Section 5. We think the novelty of this work comes from both the identification of this long-standing but underexplored problem, and the simple algorithm BIDS we propose to effectively address it.
>
> Next, we demonstrate the difference of BIDS than the MMR algorithm. MMR tries to balance relevance and diversity in the context of document retrieval, by using the following formula:
>
> $$ \text{MMR} \triangleq \arg \max_{D_i \in R - S} [ \lambda(\text{Sim}_1({D_i, Q}) - (1 - \lambda) \max_{D_j \in S} \text{Sim}_2(D_i,D_j)) ] $$
>
> , where $Q$ is a query, $D_i$ is the document that is being considered in the current iteration, and $D_j$ is a document in the subset of already selected documents, $S$. We point out that BIDS bears two major differences from MMR, despite both methods trying to optimize a form of “maximal marginal relevance”:
>
> BIDS first applies an instance-level normalization technique that is well motivated by the exploratory analysis in Section 3.
> In the iterative selection process, BIDS does not separate the query and the already selected subset. Instead, it unifies the computation of both quality and diversity a training point brings to the selected subset by comparing its influence distribution with the average influence distribution of the selected subset in an instance-level manner with regard to each validation instance. This unification is motivated by Section 3, and shows its consistent superiority to the other baseline algorithms. Due to this unification, BIDS is also exempt from the tuning of the additional parameter $\lambda$.
>
> And through ablation analysis in Section 6, we show that both of the two points above contribute positively to the improvement of final performance, and how they possibly contribute by improving the influence distribution of selected data.

---

> ### Author Response · Authors · 2024-12-04
> **Further Response to Reviewer XLPB (2/2)**
>
> ## 2. Efficiency Comparison with other data selection methods
>
> Thanks to the reviewer for raising the suggestion of more specific and exact efficiency comparison. We present the comparison of latency and memory costs between BIDS and RDS in the following table. Since in our implementation, BIDS and RDS use the same warmed-up model to compute their respective information (i.e., gradients or last-layer representations), the difference in efficiency mainly comes from the step of information computation instead of warming-up. Therefore, we only present the respective latency and GPU memory costs of the information computation step, with Llama-3-8B being the warmed-up model. Both methods use one H100 80GB GPU.
>
> |          | **Latency** | **GPU Memory** |
> |:--------:|:-----------:|:--------------:|
> | **BIDS** |   36 hours  |     44.2 GB    |
> |  **RDS** |   11 hours  |     53.6 GB    |
>
> As is shown above, since BIDS requires backpropagation to compute gradients, it has higher latency than RDS. But on the other hand, since it only computes LoRA gradients and employs an efficient random projection to reduce dimensions (which were both introduced in the original LESS work), its GPU memory costs are lower than RDS. Moreover, we would like to emphasize that the numbers listed for BIDS above are shared by all data selection methods that use the influence estimation procedure introduced by LESS. And the additional overhead incurred by the normalization and iterative selection techniques of BIDS is minimal compared with the overhead of influence estimation -- as is displayed in Appendix A.6, non-influence latency of BIDS is smaller than 1 minute with GPU acceleration, and non-influence memory cost is 0.753 GB.
>
> Finally, we want to clarify that the major goal of this work is not to reduce the overhead of current influence estimation methods, but rather identify and improve the problem of imbalanced influence for multi-task learning. We think the consistent effectiveness of BIDS across models and budgets can mostly justify the computational costs of influence estimation (which is shared by all influence-based data selection algorithms), and the low cost of the two techniques introduced by BIDS further makes it a lightweight plug-and-play module.

---

### Official Review · Reviewer_UUAb · 2024-11-01

**Soundness:** 2
**Presentation:** 3
**Contribution:** 2
**Rating:** 3
**Confidence:** 4

**Summary:**

This paper proposes a new data selection method for instruction tuning of LLMs to achieve better performance while using less data. The method proposed is BIDS, which normalizes influence scores per task and iteratively selects the most influential data for underrepresented capabilities, ensuring a balanced selection across different tasks. Experiments on LLama3 show that BIDS achieves better performance on UltraInteract benchmark over several baselines.

**Strengths:**

1. The writing is clear and easy to follow.
2. After using the proposed data selection algorithm, the performance is better than using all data on some benchmark.

**Weaknesses:**

1. The algorithm design lacks enough novelty. It simply modifies the original LESS algorithm by adding a instance-level normalization to the attribution matrix.
2. The paper does not well explain why using a balanced data selection can leads to better performance. And why using less data can leads to better performance than using all data for training. (Why excluding less "Influence" data can leads to better performance.)
3. The algorithm needs to maintain a |d| &times; |V| matrix, which is not efficiency when the dataset is large. The computation cost of the proposed algorithm needs to be discussed in the paper.
4. Authors should compare their results with other SOTA data selection algorithms to prove their effectiveness, instead of just comparing different variant of Influence-based instruction tuning data selection.

**Questions:**

What is the ratio for dividing the data into training, validation, and test sets?

---

> ### Author Response · Authors · 2024-11-22
> **Response to Reviewer UUAb (1/1)**
>
> Thank you for your valuable feedback! We appreciate that you acknowledge the presentation of our work, as well as the effectiveness of BIDS-selected data. We address your questions and concerns below.
>
> ## 1. Lack of novelty due to the simplicity of BIDS algorithm
> We kindly suggest referring to the general response, **2. Elaboration on the novelty and contribution of BIDS**, where we elaborate on why we think the simplicity of techniques used in BIDS should be treated as strength instead of weakness of this work.
>
> ## 2. Why can training on balanced data lead to better performance?
> It is a general phenomenon in LLM instruction tuning that the model’s improvement on certain tasks comes at the cost of another [1, 2, 3]. In other words, the acquisition of certain capabilities usually conflicts with others. For example, if we only scale up the amount of data targeting complicated math problem solving, then the general instruction following capability would probably be severely degraded, as is the case of full dataset training in Table 2 of the main text. Therefore, selecting a subset of training data that can balance multiple diverse capabilities and reduce the internal conflicts is a promising way to improve performance.
>
> ## 3. Why can training on less data lead to better performance than full data training?
>
> 1. According to prior research [1, 4], using all of the training data can hurt the development of specific capabilities, which is introduced in Point 2 above.
> 2. Since UltraInteract is mostly composed of synthetic data, which is known for spurious correlations, quality issues and potential biases (which were also observed in some of our early explorations), data selection is especially suitable for removing redundancy and improving the overall quality of such synthetic datasets.
>
> ## 4. The discussion on computational cost of BIDS
> We sincerely thank the reviewer for pointing out the necessity of this discussion. Below we show why BIDS does not incur much memory or latency overhead. In our training and evaluation setup, the $|D|$ dimension for the attribution matrix (AM) is about 288K, and the $|V|$ dimension is 350. Therefore, the memory cost for storing the AM using FP64 precision is less than 800MB. The latency cost for running the whole BIDS algorithm is less than 1 minute with GPU CUDA acceleration. More generally, since most of the popular mixtures of instruction finetuning data are maintained on the scale of hundreds of thousands [1, 2, 3, 5], the memory and latency cost of BIDS should be light for most of the practical training setups. We have updated this discussion of computational cost in Appendix A.6.
>
> ## 5. The comparison with other SOTA data selection baselines
> Though we regard the major goal of this work as identifying and improving influence-based data selection methods instead of proposing a completely new method from scratch, we fully understand the reviewer’s concern on comparing BIDS with non-influence-based data selection approaches. Please refer to the general response, **3. Comparison with non-influence-based data selection baseline**, where we follow the original experimental setup in the LESS paper and compare BIDS with RDS (Representation-based Data Selection).
>
> ## 6. The ratio of training/validation/test split
> As is stated in Section 3 of the main text, UltraInteract consists of the following four training subsets:
>
> - Coding: 114K
> - Logic: 11K
> - Math-CoT: 78K
> - Math-PoT: 83K
>
> As for the seven evaluation benchmarks, the validation split of each consists of 50 examples sampled from the same distribution. The ratios of validation and test splits are shown below.
>
> - HumanEval: 50/114
> - MBPP: 50/349
> - GSM-Plus: 50/10502
> - MATH: 50/4950
> - BBH: 50/6511
> - MMLU: 50/18721
> - IFEval: 50/491
>
> [1] Wang et al., How Far Can Camels Go? Exploring the State of Instruction Tuning on Open Resources, 2023
>
> [2] Ivison et al., Camels in a Changing Climate: Enhancing LM Adaptation with Tulu 2, 2023
>
> [3] Yuan et al., Advancing LLM Reasoning Generalists with Preference Trees, 2024
>
> [4] Xia et al., Using LESS Data to Tune Models (https://www.cs.princeton.edu/~smalladi/blog/2024/04/04/dataselection/), 2024
>
> [5] Yue et al., MAmmoTH: Building Math Generalist Models through Hybrid Instruction Tuning, 2023

---

> ### Author Response · Authors · 2024-11-28
> **Looking forward to further discussions**
>
> Dear Reviewer UUAb,
>
> Thank you for your valuable feedback on our submission! We have attempted to respond to your comments in our rebuttal, and tried to address your concerns with additional experiments and elaboration. We would greatly appreciate any further clarifications or discussions to enhance our work.
>
> Look forward to further discussions, and thanks again for your time and efforts!

---

### Official Review · Reviewer_kszn · 2024-11-02

**Soundness:** 2
**Presentation:** 2
**Contribution:** 2
**Rating:** 3
**Confidence:** 4

**Summary:**

This paper studies selecting suitable training data for supervised instruction fine-tuning (SFT) of pretrained LLMs. Specifically, this paper focuses on the strategy of iterative balanced selection based on the influence score between training and validation samples, named BIDS. BIDS first normalizes influence scores per task and iteratively selects training data that maximally enhances underrepresented capabilities. They provide several analyses and ablations to validate the effectiveness of core components and motivations of their approach.

**Strengths:**

Pointing out the necessity of balanced sampling for recent instruction tuning datasets, which usually consist of multiple distinctive tasks.

**Weaknesses:**

The reviewer agrees that the balanced selection should be meaningful for the multitask instruction tuning dataset.

However, the observations and discussions provided in the paper seem to be limited and less rigorous than the reviewer expected.

1. limited contribution: I understand this work as a LESS method, but I draw each sample iteratively for balance, which sounds more like an ad-hoc engineering technique for LESS. Evaluation with a single LLM backbone and single 'meaningful' baseline doesn't give me meaningful insights throughout the paper.

2. limited analysis: The most straightforward setting for balanced sampling should be to select the same number of influential samples of each training dataset (i.e., task) from the corresponding validation set (i.e., task-specific LESS selection). However, there is no comparison or discussion. Also, the impact of influence score from validation sets from a task to training samples from other tasks is not discussed, which may provide further insights on balanced sampling for underrepresented tasks.

3. marginal performance improvement: performance improvement is limited. For example, in Table 2, under a 5% budget, there is a 0.4% improvement from simple *sum*. Under 15% sample budget, it also shows 0.4% improvement against *random*. Considering the randomness of selecting validation samples and stochastic training, it looks almost like random fluctuation within the variance.  In addition, in Table 1, I wonder why the gap between the random selection and LESS has a marginal performance difference, which is quite hard to understand considering the report of the original LESS paper showing significant performance gain from random selection. This paper uses LLAMA3 (LESS uses llama2-7B, llama2-13B, and mistral-7B), but not sure this is the only reason for this.

4. limited experiments: this paper uses only one LLM backbone (llama3-7B) and uses only one meaningful baseline - LESS. More extensive comparison under the various LLM backbones following LESS paper and including various types of data pruning/selection papers, including a balanced or density-based selection approach, should be needed.

**Questions:**

Please see the weakness section.

---

> ### Author Response · Authors · 2024-11-22
> **Response to Reviewer kszn (1/2)**
>
> Thank you for your valuable suggestions! We appreciate that you acknowledge the significance of balanced data selection for multi-task instruction tuning, and address your questions and concerns below.
>
> ## 1. Limited contribution due to the simplicity of our method
> We kindly suggest referring to the general response, **2. Elaboration on the novelty and contribution of BIDS**, where we elaborate more on the contribution of BIDS to other influence-based data selection methods than LESS. We also state the reason why we think the simplicity of techniques used in BIDS should be treated as strength instead of weakness of this work.
>
> ## 2. Limited analysis
> We thank the reviewers for suggesting another balanced selection baseline of selecting an equal number of data targeting each validation task. We’ll carry out this experiment and present the comparison of results at our earliest convenience.
>
> Besides, we also appreciate the reviewer’s suggestion on discussing the influence from training examples in other tasks to a specific validation task. To make it more concrete, we analyze the “task-level influence pattern” from four subsets covered in training data (Coding, Logic, Math-CoT, Math-PoT) to the seven validation tasks. The task-level influence is computed by taking an average over all instance-level influence between examples from the specific training subset and validation task, i.e., $\text{Influence}(D, V) \triangleq \frac{1}{|D| \cdot |V|} \sum_{t \in D, v \in V} \text{Influence} (t, v)$. The results are shown below.
>
> | **Unit: $10^{-3}$** | **HumanEval** | **MBPP** |  **BBH** | **MMLU** | **GSM-Plus** | **MATH** | **IFEval** | **Avg Task Influence** |
> |:-------------------:|:-------------:|:--------:|:--------:|:--------:|:------------:|:--------:|:----------:|:----------------------:|
> |      **Coding**     |      -0.7     |   -0.8   | **-0.7** |   -0.5   |   **-0.5**   | **-0.6** |  **-0.4**  |        **-0.60**       |
> |      **Logic**      |      -0.8     |   -0.9   |   -1.4   |   -0.8   |     -1.7     |   -1.4   |    -0.8    |          -1.11         |
> |     **Math-CoT**    |    **-0.5**   | **-0.6** |   -0.9   | **-0.2** |     -1.4     |   -1.1   |    -0.5    |          -0.74         |
> |     **Math-PoT**    |      -0.7     |   -1.0   |   -1.3   |   -0.6   |     -0.8     |   -1.0   |    -0.8    |          -0.89         |
>
> This task-level influence pattern reveals several interesting characteristics:
>
> 1. For each validation task, either the Coding or the Math-CoT subset has the highest influence. And the Logic subset has the lowest influence on almost all the validation tasks. But this doesn’t necessarily mean there is also a kind of “influence score bias” across different training subsets. While influence across different validation tasks should be on the same scale for fair comparison and data selection, it’s generally natural for the influence of different training subsets on the same validation task to differ in scale – a training subset would naturally have lower influence if its answer styles or required capabilities are unmatched with the target task, thus resulting in weaker learning boosts and eventually lower influence values.
> 2. For the two coding tasks, HumanEval and MBPP, the training subset of Math-CoT has the highest influence (instead of Coding). The same pattern goes for the two mathematical tasks, GSM-Plus and MATH. But we think it can be explained by the fact that the seven validation tasks used here are mostly OOD for the training data sources in UltraInteract [1]. For example, the Coding subset is mostly synthesized based on competition-level algorithmic problems, such as CodeContest [2] and TACO [3], in contrast to basic code completion in HumanEval and MBPP. So the great discrepancy in problem difficulty and style may account for this interesting task-level influence pattern.
>
> Overall, we think it’s possible for this task-level influence distribution to provide more insight for balanced data selection on the basis of BIDS. We leave more systematic exploration in this direction to future work.
>
> [1] Yuan et al., Advancing LLM Reasoning Generalists with Preference Trees, 2024
>
> [2] Li et al., Competition-Level Code Generation with AlphaCode. 2022
>
> [3] Li et al., TACO: Topics in Algorithmic COde generation dataset, 2023

---

> ### Author Response · Authors · 2024-11-22
> **Response to Reviewer kszn (2/2)**
>
> ## 3. Limited Performance Improvement and why LESS fails to outperform random
> We acknowledge that the performance gain of training on BIDS-selected data for two epochs under the 15% budget is not very obvious. But if we further scale the training to four epochs, then 15% BIDS-selected data do beat the costly full training with a more balanced performance distribution, which, in our opinion, can make up for the previous insignificance. In addition, under the 5% and 10% budget, we both achieve marked improvements (1.3% and 1.1% respectively in macro average) compared with the random baseline, which is unlikely to be the product of stochastic errors. The same holds true for our experiments on Mistral-v0.3. We would also like to emphasize that BIDS consistently outperforms all the other baseline selection algorithms under multiple budgets and different models. Such a consistent improving trend further consolidates the effectiveness of BIDS against potential randomness in training.
>
> We are also happy to discuss more on why the naive LESS-selected data fail to outperform the random baseline. Apart from change of the model, it’s noteworthy that the original selection algorithm of LESS is optimized for “targeted instruction tuning” [4], where the evaluation is targeted towards data that embody only one specific capability and share similar reasoning and formatting patterns, such as MMLU only, BBH only or TYDIQA only. However, for the evaluation suite employed in BIDS, there are significant discrepancies in question styles and required capabilities, and we show that the original algorithm of LESS fails to balance the sampling of these divergent tasks, thus leading to suboptimal performance in macro average.
>
> ## 4. Limited experiments of base model and data selection methods
> Please refer to the general response, **1. Validating the generalizability of BIDS with different base models**, where we provide a comprehensive evaluation of BIDS on the Mistral-7B-v0.3 backbone.
>
> As for comparing with other types of data selection methods, we also kindly suggest referring to the general response, **3. Comparison with non-influence-based data selection baseline.**
>
> [4] Xia et al., LESS: Selecting Influential Data for Targeted Instruction Tuning, **Paragraph 2 in Section 1: Introduction**., 2024

---

> ### Author Response · Authors · 2024-11-28
> **Looking forward to further discussions**
>
> Dear Reviewer kszn,
>
> Thank you for your valuable feedback on our submission! We have attempted to respond to your comments in our rebuttal, and tried to address your concerns with additional experiments and elaboration. We would greatly appreciate any further clarifications or discussions to enhance our work.
>
> Look forward to further discussions, and thanks again for your time and efforts!

---

### Official Review · Reviewer_6hJ9 · 2024-11-04

**Soundness:** 3
**Presentation:** 3
**Contribution:** 3
**Rating:** 8
**Confidence:** 3

**Summary:**

The paper introduces BIDS (Balanced Influence-based Data Selection), an algorithm for improving the balance and performance of LLMs across diverse tasks during supervised instruction tuning. Traditional influence-based methods often skew data selection toward tasks with inherently higher influence scores, leading to unbalanced model capabilities. BIDS addresses this by normalizing influence values at the instance level and applying an iterative selection process that prioritizes data for underrepresented tasks. Experimental results show that BIDS achieves more balanced performance across tasks like coding, logic, math, and instruction-following.

**Strengths:**

1. strong results with only 15% of the dataset, keeping up with full-dataset training, demonstrating its potential for resource-efficient tuning of LLMs.
2. Comprehensive across diverse tasks (coding, logic, math, and instruction-following), showing consistent improvements over baseline methods, which strengthens the validity of its balanced selection approach.

**Weaknesses:**

The paper primarily focuses on specific tasks and one model (Llama-3-8B). Testing BIDS on a broader range of models (different families and sizes) and larger task sets would help assess its scalability and generalization.

**Questions:**

How robust is BIDS to noise or inaccuracies in influence score estimation? Could variations in influence scoring across tasks impact the balance achieved, and have any measures been taken to address potential score biases?

---

> ### Author Response · Authors · 2024-11-22
> **Response to Reviewer 6hJ9**
>
> We are encouraged that the reviewer acknowledged the strong results of BIDS, as well as its effect in promoting balanced data selection. Thanks for your comments and suggestions on this work! We now respond to the questions and concerns of the reviewer.
>
> ## 1. Assess the scalability and generalization of BIDS
> Please refer to the general response, **1. Validating the generalizability of BIDS with different base models**, where we provide a comprehensive evaluation of BIDS on the Mistral-7B-v0.3 model. The results show the effectiveness of BIDS across base models in different model families, thus validating its generalizability.
>
> ## 2. Robustness of BIDS to noise or inaccuracies in influence estimation process
> This is a great question. We would like to respond in two aspects:
>
> 1. The variability in the influence estimation process of LESS only has limited impact on the final performance of selected data. This statement is supported by the experimental results in the original LESS paper [1], where for each LESS-based data selection experiment, the authors of LESS conducted three trials with distinct random seeds for warmup training, and eventually found that the standard deviation of downstream performance was in a mild range on par with that of the random selection performance.
> 2. The selection process of BIDS is carried out under high numerical precision. Throughout the normalization and iterative selection, all the operations on the attribution matrix are carried out under FP64. This criterion further defends BIDS against variabilities caused by numerical errors. Also, through the prevailing results of BIDS across multiple selection budgets and model families, it’s empirically confirmed that the balance achieved by BIDS isn’t obviously impacted by the noise or inaccuracies in the influence estimation process.
>
> [1] Xia et al., LESS: Selecting Influential Data for Targeted Instruction Tuning., Table 2, Appendix A.2., 2024

---

### Author Response · Authors · 2024-11-22
**General Response (1/3)**

We thank all the reviewers for their constructive feedback! A revision of the original paper has been uploaded, where we fix typos and update additional experiments and discussions in response to the reviewers’ questions and concerns. Most of the revisions are highlighted in blue for better presentation, except for the revision of mild typos or formatting issues in tables/figures/algorithms that don’t change the meaning of the original content. Here we specifically address some of the shared concerns and questions.


## 1. Validating the generalizability of BIDS with different base models (Appendix A.5)

All the four reviewers highlighted the need for validating the effectiveness of BIDS on other base models than Llama-3-8B. Our revision compares BIDS with other baseline data selection algorithms using Mistral-7B-v0.3 as the backbone for both selection and training. The additional results are updated in Appendix A.5, and also shown below.

| **Budget** |    **Method**   | **Coding** |          | **Logic** | **STEM** | **Math** |          | **Ins-Following** | **Macro Avg** |
|:----------:|:---------------:|:----------:|:--------:|:---------:|:--------:|:--------:|:--------:|:-----------------:|:-------------:|
|            |                 |  HumanEval |   MBPP   |    BBH    |   MMLU   | GSM-Plus |   MATH   |       IFEval      |               |
|     5%     |      Random     |    36.8    |   44.3   |  **59.5** |   61.7   |   37.0   | **19.9** |        22.2       |      40.2     |
|            |   Unnormalized  |    33.3    | **45.0** |    59.3   |   61.6   |   38.0   |   18.7   |        22.0       |      39.7     |
|            |    Normalized   |    36.8    |   44.1   |    59.1   |   61.5   | **38.2** |   19.6   |      **27.5**     |    **41.0**   |
|            |       BIDS      |  **37.7**  |   44.4   |  **59.5** | **61.8** |   38.0   |   19.8   |        26.1       |    **41.0**   |
|     10%    |      Random     |    37.7    |   44.8   |    59.8   | **61.8** |   40.0   | **21.2** |        22.0       |      41.0     |
|            |   Unnormalized  |    36.0    |   43.8   |    59.7   |   61.5   | **41.6** |   20.8   |        24.6       |      41.1     |
|            |    Normalized   |    37.7    |   45.0   |    59.7   |   61.6   |   40.2   |   20.2   |        26.7       |      41.6     |
|            |       BIDS      |  **40.4**  | **46.1** |  **60.5** |   61.7   |   40.5   |   21.0   |      **27.1**     |    **42.5**   |
|     15%    | BIDS (epochs=4) |    40.4    |   47.0   |  **58.9** | **61.1** |   44.1   |   23.5   |      **28.1**     |    **43.3**   |
|    100%    | Full (epochs=4) |    41.2    |   49.3   |    54.6   |   59.4   |   48.1   |   30.1   |        19.6       |      43.2     |

We would like to highlight three observations from these results:

1. For Mistral-v0.3 under different selection budgets, BIDS still outperforms all other three baselines with a better macro-avg and more balanced task-specific performance.
2. Specifically, the improvements of Unnormalized → Normalized → BIDS are consistent with our prior observation on Llama-3-8B (Section 6 of main text). We also carried out the same set of analysis of the influence distribution of selected data (Section 6.2 of main text) by probing into AID and THI metrics. These analysis figures are presented in Appendix A.5. Both AID and THI metrics suggest a balancing trend, though the degree of the original imbalance in influence scores for Mistral-v0.3 is not as high as Llama-3.
3. Notably, the top 15% subset selected by BIDS again outperforms full dataset training in macro average. With the increase of the scale of training data, BIDS not only continues to improve its performance on Coding and Math, but also maintains its remarkable instruction-following ability, which is severely degraded in the case of full dataset training.

These results confirm the generalizability of BIDS to different base models, especially given that Llama-3 and Mistral-v0.3 belong to different backbone families, and thus probably have divergent pre-training data compositions.

---

> ### Author Response · Authors · 2024-11-22
> **General Response (2/3)**
>
> ## 2. Elaboration on the novelty and contribution of BIDS
>
> Reviewer kszn, UUAB and XLPB expressed their concern about the novelty and contribution of BIDS. Two points were mainly proposed:
>
> 1. The empirical experiments of BIDS heavily rely on the prior work LESS, which may undermine the scope of its contribution.
> 2. The technical design of BIDS is relatively simple, which suggests its limited technical contribution and novelty.
>
> We thank the reviewers for raising these two fundamental points, and would like to elaborate more on the motivation of BIDS, so that its contribution and novelty can be better justified.
>
> ### 2.1 Elaboration on Point 1
> As is stated in Section 2 of the main text, BIDS is motivated by a common problem formulation shared by multiple LLM-scale gradient-based influence estimation methods ([1], [2], [3]). They all need a validation set to represent a targeted data distribution that the selected data are optimized for, and require the computation of pointwise data influence between each training instance and the validation data. In this work, we aim to specifically extend such an influence-based data selection framework to a multi-capability instruction tuning setup, which were rarely explored in prior research. And we ground our empirical study on LESS mainly because it’s the only method that directly targets instruction tuning among these three LLM-scale approaches. But this practical grounding doesn’t hurt the following two properties of BIDS:
>
> 1. The normalization and selection techniques introduced by BIDS are fully compatible with all these gradient-based influence estimation methods. BIDS can serve as a plug-and-play module that induces little additional cost [detailed in our response to reviewer UUAb].
> 2. Due to the essential similarity in influence modeling patterns among these methods, the results of our work should also provide useful and reliable insight for other influence-based data selection approaches beyond instruction finetuning for LLMs.
>
> Therefore, BIDS follows a well-established influence estimation pattern adopted by many previous works including but not limited to LESS. While we focus on instruction tuning, we conjecture that BIDS is equally applicable to other influence-based data selection approaches for both instruction tuning and pretraining, and provides great insights into the behavior of SOTA data selection methods in multi-capability learning.
>
> ### 2.2 Elaboration on Point 2
>
> Research can be novel in multiple ways. Some propose new methods, while others provide fresh insights that deepen our understanding or uncover previously overlooked phenomena. In our opinion, both are equally valuable and contribute to scientific progress in complementary ways.
>
> To the best of our knowledge, our findings on the limitations of existing influence-based data selection methods in multi-capability learning settings are novel, as is our analysis framework to identify these issues. To address them, we intentionally choose to use simple, well-established techniques over more complex alternatives because they are time-tested and robust in practice. Moreover, these techniques are also well motivated by the systematic exploratory analysis specific to our training and evaluation setup. Therefore, we believe our scientific findings should be considered alongside the BIDS algorithm when evaluating the novelty of this work; our decision of using simple and proven techniques should be considered a strength rather than a weakness.
>
> [1] Xia et al., LESS: Selecting Influential Data for Targeted Instruction Tuning, 2024
> [2] Choe et al., What is Your Data Worth to GPT? LLM-Scale Data Valuation with Influence Functions, 2024
> [3] Yu et al., MATES: Model-Aware Data Selection for Efficient Pretraining with Data Influence Models, 2024

---

> ### Author Response · Authors · 2024-11-22
> **General Response (3/3)**
>
> ## 3. Comparison with non-influence-based data selection baseline (Section 5)
>
> Reviewer kszn, UUAB and XLPB uniformly suggested comparing BIDS with other non-influence-based data selection methods. We sincerely appreciate this great advice, as we also consider it necessary to show that influence-based data selection, for which BIDS is designed, is a class of state-of-the-art data selection approaches that is indeed worth investigating and further optimizing.
>
> As for the specific non-influence selection method, we follow the same experimental setup in the original LESS paper and opt for **RDS (Representation-based Data Selection [4][5])**. RDS uses the language model’s hidden representation for data selection. More concretely, it extracts the final layer representation of the last token of each example sequence, and computes the cosine similarity scores between training and validation examples. Training examples with the highest similarity to any one of the validation examples are selected. To ensure fair comparison, we use the same Llama-3-8B model that already goes through the warmup step for BIDS to extract the final layer 4096-dimensional representations. The comparison results between RDS and BIDS are shown below and have also been updated in Section 5 of the main text.
>
> | **Budget** | **Method** | **Coding** |          | **Logic** | **STEM** | **Math** |          | **Ins-Following** | **Macro Avg** |
> |:----------:|:----------:|:----------:|:--------:|:---------:|:--------:|:--------:|:--------:|:-----------------:|:-------------:|
> |            |            |  HumanEval |   MBPP   |    BBH    |   MMLU   | GSM-Plus |   MATH   |       IFEval      |               |
> |     5%     |   Random   |    43.5    |   48.9   |  **64.8** | **64.9** |   41.5   |   22.5   |        18.1       |      43.4     |
> |            |     RDS    |  **45.6**  | **52.7** |    62.2   |   65.0   |   34.5   |   17.2   |        15.5       |      41.8     |
> |            |    BIDS    |  **45.6**  |   51.0   |    64.3   | **64.9** | **42.1** | **22.9** |      **21.4**     |    **44.6**   |
> |     10%    |   Random   |    47.8    |   50.6   |    65.0   | **64.9** |   43.9   |   24.0   |        17.8       |      44.9     |
> |            |     RDS    |  **50.0**  | **54.7** |    63.2   |   64.6   |   39.3   |   22.4   |        18.3       |      44.6     |
> |            |    BIDS    |    48.2    |   50.4   |  **65.1** | **64.9** | **45.1** | **25.1** |      **23.4**     |    **46.0**   |
> |     15%    |   Random   |    48.7    |   51.9   |  **65.2** | **65.1** |   45.6   |   25.0   |        18.8       |      45.7     |
> |            |     RDS    |  **50.0**  | **53.9** |    63.7   |   64.5   |   41.1   |   23.5   |        18.1       |      45.0     |
> |            |    BIDS    |    49.1    |   50.7   |    63.7   |   64.6   | **45.8** | **26.2** |      **22.6**     |    **46.1**   |
>
> As can be seen in the table, the performance of RDS-selected data is significantly biased towards the two coding tasks, HumanEval and MBPP. However, under all the three budgets, the uniformly better performance of RDS on coding tasks comes at the cost of severely degraded performance on almost all other tasks. Specifically, RDS shows the most significant performance gap in math and instruction-following, outperformed by the random baseline on almost all of these results. These observations confirm the value of further improving influence-based data selection methods in the multi-capability learning setup. More interestingly, they also suggest the possibility that the imbalance of “utility scores [6]” may exist for both influence- and non-influence-based data selection approaches. We leave detailed investigations into the more general imbalance of utility scores to future work.
>
> [4] Zhang et al., The Unreasonable Effectiveness of Deep Features as a Perceptual Metric, 2018
>
> [5] Hanawa et al., Evaluation of Similarity-based Explanations, 2020
>
> [6] Yin and Rush., Compute-Constrained Data Selection, 2024

---

### Note · Authors · 2024-12-16

I have read and agree with the venue's withdrawal policy on behalf of myself and my co-authors.